# Leveraging the two-timescale regime
# to demonstrate convergence of neural networks

**Pierre Marion**
Sorbonne Université, CNRS,
Laboratoire de Probabilités, Statistique et Modélisation, LPSM,
F-75005 Paris, France
`pierre.marion@sorbonne-universite.fr`

**Raphaël Berthier**
EPFL, Switzerland
`raphael.berthier@epfl.ch`

## Abstract

We study the training dynamics of shallow neural networks, in a two-timescale regime in which the stepsizes for the inner layer are much smaller than those for the outer layer. In this regime, we prove convergence of the gradient flow to a global optimum of the non-convex optimization problem in a simple univariate setting. The number of neurons need not be asymptotically large for our result to hold, distinguishing our result from popular recent approaches such as the neural tangent kernel or mean-field regimes. Experimental illustration is provided, showing that the stochastic gradient descent behaves according to our description of the gradient flow and thus converges to a global optimum in the two-timescale regime, but can fail outside of this regime.

## 1 Introduction

Artificial neural networks are among the most successful modern machine learning methods, in particular because their non-linear parametrization provides a flexible way to implement feature learning (see, e.g., Goodfellow et al., 2016, chapter 15). Following this empirical success, a large body of work has been dedicated to understanding their theoretical properties, and in particular to analyzing the optimization algorithm used to tune their parameters. It usually consists in minimizing a loss function through stochastic gradient descent (SGD) or a variant (Bottou et al., 2018). However, the non-linearity of the parametrization implies that the loss function is non-convex, breaking the standard convexity assumption that ensures global convergence of gradient descent algorithms.

In this paper, we study the training dynamics of *shallow* neural networks, i.e., of the form

$$f(x; a, u) = a_0 + \sum_{j=1}^{m} a_j g(x; u_j)\,,$$

where $m$ denotes the number of hidden neurons, $a = (a_0, \ldots, a_m)$ and $u = (u_1, \ldots, u_m)$ denote respectively the outer and inner layer parameters, and $g(x; u)$ denotes a non-linear function of $x$ and $u$. The novelty of this work lies in the use of a so-called *two-timescale regime* (Borkar, 1997) to train the neural network: we set stepsizes for the inner layer $u$ to be an order of magnitude smaller than the stepsizes of the outer layer $a$. This ratio is controlled by a parameter $\varepsilon$. In the regime $\varepsilon \ll 1$, the neural network can be thought of as a fitted linear regression with slowly evolving features

37th Conference on Neural Information Processing Systems (NeurIPS 2023).

$g(x; u_j)$, $j = 1, \ldots, m$: this reduction enables us to precisely describe the movement of the inner layer parameters $u_j$.

Our approach proves convergence of the *gradient flow* to a global optimum of the non-convex landscape with a *fixed* number $m$ of neurons. The gradient flow can be seen as the simplifying yet insightful limit of the SGD dynamics as the stepsize $h$ vanishes. Proving convergence with a fixed number of neurons contrasts with two other popular approaches that require to take the limit $m \to \infty$: the neural tangent kernel (Jacot et al., 2018; Allen-Zhu et al., 2019; Du et al., 2019; Zou et al., 2020) and the mean-field approach (Chizat and Bach, 2018; Mei et al., 2018; Rotskoff and Vanden-Eijnden, 2018; Sirignano and Spiliopoulos, 2020). As a consequence, this paper is intended as a step towards understanding feature learning with a moderate number of neurons.

While our approach through the two-timescale regime is general, our description of the solution of the two-timescale dynamics and our convergence results are specific to a simple example showcasing the approach. More precisely, we consider univariate data $x \in [0, 1]$ and non-linearities of the form $g(x; u_j) = \sigma(\eta^{-1}(x - u_j))$, where $u_j$ is a variable translation parameter, $\eta$ is a fixed dilatation parameter, and $\sigma$ is a sigmoid-like non-linearity. Finally, we restrict ourselves to the approximation of piecewise constant functions.

**Organization of this paper.**   In Section 2, we detail our setting and state our main theorem on the convergence of the gradient flow to a global optimum. Section 3 articulates this paper with related work. Section 4 provides a self-contained introduction to the *two-timescale limit* $\varepsilon \to 0$. We explain how it simplifies the analysis of neural networks, and provides heuristic predictions for the movement of neurons in our setting. Section 5 gives a rigorous derivation of our result. We prove convergence first in the two-timescale limit $\varepsilon \to 0$, then in the two-timescale regime with $\varepsilon$ small but positive. Section 6 presents numerical experiments showing that the SGD dynamics follow closely those of the gradient flow in the two-timescale regime, and therefore exhibit convergence to a global optimum. On the contrary, SGD can fail to reach a global optimum outside of the two-timescale regime.

## 2   Setting and main result

We present a setting in which a piecewise constant univariate function $f^* : [0, 1] \to \mathbb{R}$ is learned with gradient flow on a shallow neural network. Our notations are summarized on Figure 1. We begin by introducing our class of functions of interest.

**Definition 1.** *Let $n \geqslant 2$, $\Delta v \in (0, 1)$, $\Delta f > 0$ and $M \geqslant 1$. We denote $\mathcal{F}_{n,\Delta v,\Delta f,M}$ the class of functions $f^* : [0, 1] \to \mathbb{R}$ satisfying the following conditions:*

- *$f^*$ is piecewise constant: there exists*

$$0 = v_0 < v_1 < \cdots < v_{n-1} < v_n = 1$$

  *and $f_0^*, \ldots, f_{n-1}^* \in \mathbb{R}$ such that*

$$\forall x \in (v_i, v_{i+1}), f^*(x) = f_i^*,$$

- *for all $i \in \{1, \ldots, n\}, v_i - v_{i-1} \geqslant \Delta v$,*
- *for all $i \in \{1, \ldots, n-1\}, |f_i^* - f_{i-1}^*| \geqslant \Delta f$,*
- *for all $i \in \{0, \ldots, n-1\}, |f_i^*| \leqslant M$.*

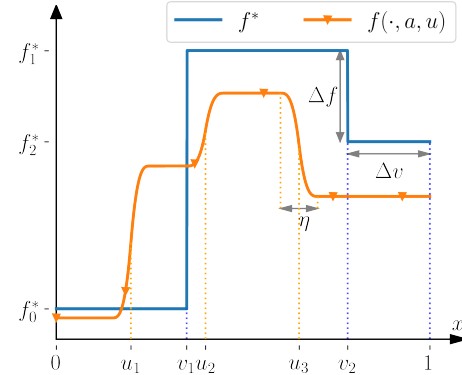

Figure 1: Notations of the paper. The target $f^*$ is in blue and the neural network $f(\cdot; a, u)$ in orange.

Let us now define our class of neural networks. Consider $\sigma : \mathbb{R} \to \mathbb{R}$ an increasing, twice continuously differentiable non-linearity such that $\sigma(x) = 0$ if $x \leqslant -1/2$, $\sigma(x) = 1$ if $x \geqslant 1/2$, and $\sigma - 1/2$ is odd. Then, our class of shallow neural networks is defined by

$$f(x; a, u) = a_0 + \sum_{j=1}^{m} a_j \sigma_\eta(x - u_j), \qquad \sigma_\eta(x) = \sigma(\eta^{-1} x),$$

where $0 < \eta \leqslant 1$ measure the sharpness of the non-linearity $\sigma_\eta$. Note that that inner layer parameter $u_j$ determines the translation of the non-linearity; no parameterized multiplicative operation on $x$ is performed in this layer. We refer to the parameter $u$ as the "positions" of the neurons (or, sometimes, simply as the "neurons") and to the parameter $a$ as the "weights" of the neurons. We define the quadratic loss as

$$L(a, u) = \frac{1}{2} \int_0^1 (f^*(x) - f(x; a, u))^2 \, \mathrm{d}x \,.$$

We use gradient flow on $L$ to fit the parameters $a$ and $u$: they evolve according to the dynamics

$$\frac{\mathrm{d}a}{\mathrm{d}t}(t) = -\nabla_a L(a(t), u(t)) \,, \qquad\qquad \frac{\mathrm{d}u}{\mathrm{d}t}(t) = -\varepsilon \nabla_u L(a(t), u(t)) \,, \qquad (1)$$

where $\varepsilon$ corresponds to the ratio of the stepsizes of the two iterations.

**Main result.** By leveraging the two-timescale regime where $\varepsilon$ is small, our theorem shows that, with high probability, a neural network trained with gradient flow is able to recover an arbitrary piecewise constant function to an arbitrary precision. The proof is relegated to the Appendix.

**Theorem 1.** *Let $\xi, \delta > 0$, and $f^*$ a piecewise constant function from $\mathcal{F}_{n,\Delta v, \Delta f, M}$. Assume that the neural network has $m$ neurons with*

$$m \geqslant \frac{6}{\Delta v} \left( 4 + \log n + \log \frac{1}{\delta} \right) . \qquad (2)$$

*Assume that, at initialization, the positions $u_1, \ldots, u_m$ of the neurons are i.i.d. uniformly distributed on $[0, 1]$ and their weights $a_0, \ldots, a_m$ are equal to zero.*

*Then there exists $Q_1 > 0$ and $Q_2 > 0$ depending on $\xi, \delta, m, \Delta f, M$ such that, if*

$$\eta \leqslant Q_1 \,, \qquad\qquad\qquad \varepsilon \leqslant Q_2 \,, \qquad (3)$$

*then, with probability at least $1 - \delta$, the solution to the gradient flow (1) is defined at least until $T = \frac{6}{\varepsilon(\Delta f)^2}$, and*

$$\int_0^1 |f^*(x) - f(x; a(T), u(T))|^2 \, \mathrm{d}x \leqslant \xi \,.$$

*Further, $Q_1 = \dfrac{C_1}{M^2(m+1)} \min \left( \dfrac{\delta^2(\Delta f)^2}{(m+1)^4}, \xi \right)$ and $Q_2 = \dfrac{C_2 \delta^2}{M^4(m+1)^{17/2}} \min \left( \dfrac{\delta(\Delta f)^2}{m+1}, \xi \right)$ for some universal constants $C_1, C_2 > 0$.*

For this result to hold, the inequality (2) requires the number of neurons in the neural network to be large enough. Note that the minimum number of neurons required to approximate the $n$ pieces of $f^*$ is equal to $n$. If the length of all the intervals is of the same order of magnitude, then $\Delta v = \Theta(1/n)$ and thus the condition is $m = \Omega(n(1 + \log n + \log 1/\delta))$. In this case, condition (2) only adds a logarithmic factor in $n$ and $\delta$. Moreover, the lower bound on $m$ does not depend on the target precision $\xi$. Thus we observe some non-linear feature learning phenomenon: with a fixed number $m$ of neurons, gradient flow on a neural network can approximate any element from the infinite-dimensional space of piecewise constant functions to an arbitrary precision.

The recovery result of Theorem 1 is provided under two conditions (3). The first one should not surprise the reader: the condition on $\eta$ enables the non-linearity to be sharp enough in order to approximate well the jumps of the piecewise constant function $f^*$. The novelty of our work lies in the condition on $\varepsilon$, that we refer to as the *two-timescale regime*. This condition ensures that the stepsizes taken in the positions $u$ are much smaller than the stepsizes taken in the weights $a$. As a consequence, the weights $a$ are constantly close to the best linear fit given the current positions $u$. This property decouples the dynamics of the two layers of the neural network; this enables a sharp description of the gradient flow trajectories and thus the recovery result shown above. This intuition is detailed in Section 4.

# 3 Related work

**Two-timescale regime.** Systems with two timescales, or *slow-fast systems*, have a long history in physics and mathematics, see Berglund and Gentz (2006, Chapter 2) for an introduction. In particular, iterative algorithms with two timescales have been used in stochastic approximation and optimization, see Borkar (1997) or Borkar (2009, Section 6). For instance, they are used in the training of generative adversarial networks, to decouple the dynamics of the generator from those of the discriminator (Heusel et al., 2017), in reinforcement learning, to decouple the value function estimation from the temporal difference learning (Szepesvári, 2010), or more generally in bilevel optimization, to decouple the outer problem dynamics from the inner problem dynamics (Hong et al., 2023). However, to the best of our knowledge, the two-timescale regime has not been used to show convergence results for neural networks.

**Layer-wise learning rates.** Practitioners are interested in choosing learning rates that depend on the layer index to speed up training or improve performance. Using smaller learning rates for the first layers and higher learning rates for the last layer(s) improves performance for fine-tuning (Howard and Ruder, 2018; Ro and Choi, 2021) and is a common practice for transfer learning (see, e.g., Li et al., 2022). Another line of work proposes to update layer-wise learning rates depending on the norm of the gradients on each layer (Singh et al., 2015; You et al., 2017; Ko et al., 2022). However, they aim to compensate the differences across gradient norms in order to learn all the parameters at the same speed, while on the contrary we enjoy the theoretical benefits of learning different speeds.

**Theory of neural networks.** A key novelty of the analysis of this paper is that we show recovery with a fixed number of neurons. We now detail the comparison with other analyses.

The neural tangent kernel regime (Jacot et al., 2018; Allen-Zhu et al., 2019; Du et al., 2019; Zou et al., 2020) corresponds to small movements of the parameters of the neural network. In this case, the neural network can be linearized around its initial point, and thus behaves like a linear regression. However, in this regime, the neural network can approximate only a finite dimensional space of functions, and thus it is necessary to take $m \to \infty$ to be able to approximate the infinite-dimensional space of piecewise constant functions to an arbitrary precision.

The mean-field regime (Chizat and Bach, 2018; Mei et al., 2018; Rotskoff and Vanden-Eijnden, 2018; Sirignano and Spiliopoulos, 2020) describes the dynamics of two-layer neural networks in the regime $m \gg 1$ through a partial differential equation on the density of neurons. This regime is able to describe some non-linear feature learning phenomena, but does not explained the observed behavior with a moderate number of neurons. In this paper, we show that in the two-timescale regime, only a single neuron aligns with each of the discontinuities of the function $f^*$. However, it should be noted that the neural tangent kernel and mean-field regimes have been applied to show recovery in a wide range of settings, while our work is restricted to the recovery of piecewise constant functions. Extending the application of the two-timescale regime is left for future work.

Our work includes a detailed analysis of the alignment of the positions of the neurons with the discontinuities of the target function $f^*$. This is analogous to a line of work (see, e.g., Saad and Solla (1995); Goldt et al. (2020); Veiga et al. (2022)) interested in the alignment of a "student" neural network with the features of a "teacher" neural network that generated the data, for high-dimensional Gaussian input. In general, the non-linear evolution equations describing this alignment are hard to study theoretically. On the contrary, thanks to the two-timescale regime and to the simple setting of this paper, we are able to give a precise description of the movement of the neurons.

Our study bears high-level similarities with the recent work of Safran et al. (2022). In a univariate classification setting, they show that a two-layer neural network achieves recovery with a number of neurons analogous to (2): inversely proportional to the length of the smallest constant interval of the target, up to logarithmic terms in the number of constant intervals and in the failure probability. However, the two papers have different settings: Safran et al. (2022) consider classification with ReLU activations while we consider regression with sigmoid-like activations. More importantly, the authors do not use the two-timescale regime. Instead, by a specific initialization scale, they ensure that the neural network has a first lazy phase where the positions of the neurons do not move significantly. For the second rich phase, they describe the implicit bias of the limiting point; this approach does not lead to an estimate of the convergence time while the fine description of the two-timescale limit does.

Finally, a related technique is the so-called layerwise training, which consists in first training the inner layer with the outer layer fixed, and then doing the reverse. This setup has been used in theoretical works to show convergence of (stochastic) gradient descent in a feature learning regime with a moderate number of neurons (Abbe et al., 2023; Damian et al., 2022). The two-timescale regime can be seen as a refinement of this technique since we allow both layers to move simultaneously instead of sequentially, which is closer to practical setups.

## 4   A non-rigorous introduction to the two-timescale limit

This section introduces the core ideas of our analysis in a non-rigorous way. Section 4.1 introduces the limit of the dynamics when $\varepsilon \to 0$, called the *two-timescale limit*. Section 4.2 applies the two-timescale limit to predict the movement of the neurons.

### 4.1   Introduction to the two-timescale limit

Let us consider the gradient flow equations (1) and perform the change of variables $\tau = \varepsilon t$:

$$\frac{\mathrm{d}a}{\mathrm{d}\tau} = -\frac{1}{\varepsilon}\nabla_a L(a,u)\,, \qquad\qquad \frac{\mathrm{d}u}{\mathrm{d}\tau} = -\nabla_u L(a,u)\,. \qquad (4)$$

In the two-timescale regime $\varepsilon \ll 1$, the rate of the gradient flow in the weights $a$ is much larger than then the rate in the positions $u$. Note that $L$ is marginally convex in $a$, and thus, for a fixed $u$, the gradient flow in $a$ must converge to a global minimizer of $a \mapsto L(a,u)$. More precisely, assume that $\{\sigma_\eta(.-u_1), \ldots, \sigma_\eta(.-u_m)\}$ forms an independent set of functions in $L^2([0,1])$. Then the global minimizer of $a \mapsto L(a,u)$ is unique; we denote it as $a^*(u)$. In the limit $\varepsilon \to 0$, we expect that $a$ evolves sufficiently quickly with respect to $u$ so that it converges instantaneously to $a^*(u)$. In other words, the gradient flow system (4) reduces to its so-called *two-timescale limit* when $\varepsilon \to 0$:

$$a = a^*(u)\,, \qquad\qquad \frac{\mathrm{d}u}{\mathrm{d}\tau} = -\nabla_u L(a^*(u),u)\,. \qquad (5)$$

The two-timescale limit considerably simplifies the study of the gradient flow system because it substitutes the weights $a$, determined to be equal to $a^*(u)$. However, showing that (4) reduces to (5) requires some mathematical care, including checking that $a^*(u)$ is well-defined.

**Remark 1** (abuse of notation). *Equation* (5) *contains an ambiguous notation: does* $\nabla_u L(a^*(u),u)$ *denote the gradient in* $u$ *of the map* $L^* : u \mapsto L(a^*(u),u)$ *or the gradient* $(\nabla_u L)(a,u)$ *taken at* $a = a^*(u)$? *In fact, by definition of* $a^*(u)$*, both quantities coincide:*

$$(\nabla_u L^*)(u) = \left(\frac{\mathrm{d}a^*}{\mathrm{d}u}(u)\right)^\top (\nabla_a L)(a^*(u),u) + (\nabla_u L)(a^*(u),u) = (\nabla_u L)(a^*(u),u)\,,$$

*where* $\frac{\mathrm{d}a^*}{\mathrm{d}u}$ *denotes the differential of* $a^*$ *in* $u$ *and we use that, by definition of* $a^*(u)$*,* $(\nabla_a L)(a^*(u),u) = 0$. *This is a special case of the envelope theorem* (Border, 2015, *Sec. 5.10*).

The discussion in this section is not specific to the setting of Section 2. Using the two-timescale limit to decouple the dynamics of the outer layer $a$ and the inner layer $u$ is a general tool that may be used in the study of any two-layer neural network. We chose the specific setting of this paper so that the two-timescale limit (5) can be easily studied, thereby showcasing the approach. The next section is devoted to a sketch of this study.

### 4.2   Sketch of the dynamics of the two-timescale limit

In this section, in order to simplify the exposition of the behavior of the two-timescale limit (5), we consider the limiting case $\eta \to 0$. Note that this is coherent with Theorem 1 that requires $\eta$ to be small. This limit is a neural network with a non-linearity equal to the Heaviside function

$$\sigma_0(x) = 0 \quad \text{if } x < 0\,, \qquad \sigma_0(x) = 1/2 \quad \text{if } x = 0\,, \qquad \sigma_0(x) = 1 \quad \text{if } x > 0\,.$$

Note that $\sigma_0$ would be a poor choice of non-linearity in practice: as its derivative is $0$ almost everywhere, the positions $u$ would not move. However, it is a relevant tool to get an intuition about the dynamics of our system for a small $\eta$. Moreover, as we will see in Section 6, the dynamics sketched here match closely those of the SGD (with $\eta > 0$).

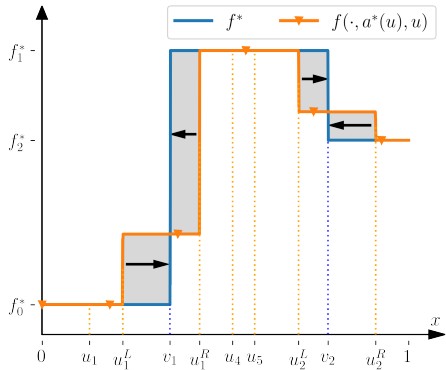

Figure 2: Sketch of the dynamics of the neurons in the two-timescale limit with a Heaviside non-linearity. Only the neurons next to a discontinuity of the target move.

The set $\{1, \sigma_0(. - u_1), \ldots, \sigma_0(. - u_m)\}$ generates the space of functions that are piecewise constant with respect to the subdivision $\{u_1, \ldots, u_m\}$. Furthermore, if $u_1, \ldots, u_m$ are distinct and in $(0, 1)$, then this set is an independent set of functions in $L^2([0, 1])$. Thus $a^*(u)$ is well defined and represents the coefficients of the best piecewise constant approximation of $f^*$ with subdivision $\{u_1, \ldots, u_m\}$.

This quantity is straightforward to describe under the mild additional assumption that there are at least two neurons $u_j$ in each interval $(v_{i-1}, v_i)$ between two points of discontinuity of $f^*$. For each $1 \leqslant i \leqslant n$, let $u_i^{\mathrm{L}}$ denote the largest position of neurons below $v_i$ and $u_i^{\mathrm{R}}$ denote the smallest position above $v_i$ (with convention $u_0^{\mathrm{R}} = 0$ and $u_{n+1}^{\mathrm{L}} = 1$). By assumption, $0 = u_0^{\mathrm{R}} < u_1^{\mathrm{L}} < u_1^{\mathrm{R}} < \cdots < u_n^{\mathrm{L}} < u_n^{\mathrm{R}} < u_{n+1}^{\mathrm{L}} = 1$ are distinct. A simple computation then shows the following identities:

- for all $i \in \{1, \ldots, n\}$, for all $x \in (u_i^{\mathrm{L}}, u_i^{\mathrm{R}})$, $f(x; a^*(u), u) = \dfrac{v_i - u_i^{\mathrm{L}}}{u_i^{\mathrm{R}} - u_i^{\mathrm{L}}} f_{i-1}^* + \dfrac{u_i^{\mathrm{R}} - v_i}{u_i^{\mathrm{R}} - u_i^{\mathrm{L}}} f_i^*$,

- and for all $i \in \{1, \ldots, n+1\}$, for all $x \in (u_{i-1}^{\mathrm{R}}, u_i^{\mathrm{L}})$, $f(x; a^*(u), u) = f_{i-1}^*$,

where we recall that $f_i^*$ denotes the value of $f^*$ on the interval $(v_i, v_{i+1})$. Figure 2 illustrates the situation. Moreover, the loss $L(a^*(u), u)$, which is half of the square $L^2$-error of this optimal approximation, can be written

$$L(a^*(u), u) = \frac{1}{2} \sum_{i=1}^{n-1} \frac{(v_i - u_i^{\mathrm{L}})(u_i^{\mathrm{R}} - v_i)}{u_i^{\mathrm{R}} - u_i^{\mathrm{L}}} (f_i^* - f_{i-1}^*)^2 . \tag{6}$$

The dynamics of the two-timescale limit (5) corresponds to the local optimization of the subdivision $u$ in order to minimize the loss (6). A remarkable property of this loss is that it decomposes as a sum of local losses around the jump points $v_i$ for $i \in \{1, \ldots, n-1\}$. Each element of the sum involves only the two neurons located at $u_i^{\mathrm{L}}$ and $u_i^{\mathrm{R}}$. As a consequence, the dynamics of the two-timescale limit (5) decompose as $n$ independent systems of two neurons $u_i^{\mathrm{L}}$ and $u_i^{\mathrm{R}}$: for all $i \in \{1, \ldots, n-1\}$,

$$
\begin{aligned}
\frac{\mathrm{d}u_i^{\mathrm{L}}}{\mathrm{d}\tau} &= -\frac{\mathrm{d}L}{\mathrm{d}u_i^{\mathrm{L}}}(a^*(u), u) = +\frac{1}{2} \frac{(u_i^{\mathrm{R}} - v_i)^2}{(u_i^{\mathrm{R}} - u_i^{\mathrm{L}})^2}(f_i^* - f_{i-1}^*)^2 , \\
\frac{\mathrm{d}u_i^{\mathrm{R}}}{\mathrm{d}\tau} &= -\frac{\mathrm{d}L}{\mathrm{d}u_i^{\mathrm{R}}}(a^*(u), u) = -\frac{1}{2} \frac{(v_i - u_i^{\mathrm{L}})^2}{(u_i^{\mathrm{R}} - u_i^{\mathrm{L}})^2}(f_i^* - f_{i-1}^*)^2 .
\end{aligned}
\tag{7}
$$

All neurons other than $u_1^{\mathrm{L}}, u_1^{\mathrm{R}}, \ldots, u_{n-1}^{\mathrm{L}}, u_{n-1}^{\mathrm{R}}$ do not play a role in the expression (6), thus they do not move in the two-timescale limit (5). The position $u_i^{\mathrm{L}}$ moves right and $u_i^{\mathrm{R}}$ moves left, until one of them hits the point $v_i$. This shows that the positions of the neurons eventually align with the jumps of the function $f^*$, and thus that the function $f^*$ is recovered.

# 5 Convergence of the gradient flow

In this section, we give precise mathematical statements leading to the convergence of the gradient flow to a global optimum, first in the two-timescale limit $\varepsilon \to 0$, then in the two-timescale regime with $\varepsilon$ small but positive. All proofs are relegated to the Appendix.

## 5.1 In the two-timescale limit

This section analyses rigorously the two-timescale limit (5), which we recall for convenience:

$$a^*(u) = \arg\min_a L(a, u), \qquad \frac{\mathrm{d}u}{\mathrm{d}\tau} = -\nabla_u L(a^*(u), u). \tag{8}$$

We start by giving a rigorous meaning to these equations. First, for $L$ to be differentiable in $u$, we require the parameter $\eta$ of the non-linearity to be positive. Second, for $a^*(u)$ to be well-defined, we need $u \mapsto L(a, u)$ to have a unique minimum. Obviously, if the $u_i$ are not distinct, then the features $\{\sigma_\eta(. - u_1), \ldots, \sigma_\eta(. - u_m)\}$ are not independent and thus the minimum can not be unique. We restrict the state space of our dynamics to circumvent this issue. For $u \in [0, 1]^m$, we denote

$$\Delta(u) = \min_{0 \leqslant j, k \leqslant m+1, \, j \neq k} |u_j - u_k|,$$

with the convention that $u_0 = -\eta/2$ and $u_{m+1} = 1 + \eta/2$. Further, we define $\mathcal{U} = \{u \in [0, 1]^m \,|\, \Delta(u) > 2\eta\}$. The proposition below shows that $\mathcal{U}$ gives a good candidate for a set supporting solutions of (8).

**Proposition 1.** *For $u \in \mathcal{U}$, the Hessian $H(u)$ of the quadratic function $L(., u)$ is positive definite and its smallest eigenvalue is greater than $\Delta(u)/8$. In particular, $L(., u)$ has a unique minimum $a^*(u)$.*

The bound on the Hessian is useful in the following, in particular in the proof of the following result.

**Proposition 2.** *Let $G(u) = \nabla_u L(a^*(u), u)$ for $u \in \mathcal{U}$. Then $G : \mathcal{U} \to \mathbb{R}^m$ is Lipschitz-continuous.*

Then, the Picard-Lindelöf theorem (see, e.g., Luk, 2017 for a self-contained presentation and Arnold, 1992 for a textbook) guarantees, for any initialization $u(0) \in \mathcal{U}$, the existence and unicity of a maximal solution of (8) taking values in $\mathcal{U}$. This solution is defined on a maximal interval $[0, T_{\max})$ where it could be that $T_{\max} < \infty$ if $u$ hits the boundary of $\mathcal{U}$. However, the results below show that the target function $f^*$ is recovered before this happens (with high probability over the initialization), and thus that this notion of solution is sufficient for our purposes. To this aim, we first define some sufficient conditions that the initialization should satify.

**Definition 2.** *Let $D$ be a positive real. We say that a vector of positions $u \in [0, 1]^m$ is $D$-good if*

  *(a) for all $i \in \{0, \ldots, n-1\}$, there are at least 6 positions $u_j$ in each interval $[v_i, v_{i+1}]$,*

  *(b) $\Delta(u) \geqslant D$, and*

  *(c) for all $i \in \{1, \ldots, n-1\}$, denoting $u_i^{\mathrm{L}}$ the position closest to the left of $v_i$ and $u_i^{\mathrm{R}}$ the position closest to the right, we have $|u_i^{\mathrm{R}} + u_i^{\mathrm{L}} - 2v_i| \geqslant D$.*

Condition (a) is related to the fact that the derivation in Section 4.2 is valid only if there are at least two neurons per piece. This requirement that the neurons be distributed on every piece of the target seems to be necessary for our result to hold, and we provide in Appendix D.1 a counter-example where recovery fails otherwise. Condition (b) indicates that the neurons have to be sufficiently spaced at initialization, which is not surprising since we have to guarantee that $\Delta(u(\tau)) > 2\eta$, that is, $u(\tau) \in \mathcal{U}$, for all $\tau$ until the recovery of $f^*$ happens. Finally, condition (c) also helps to control the distance between neurons: although $u_i^{\mathrm{L}}$ and $u_i^{\mathrm{R}}$ move towards each other, as shown by (7), their distance can be controlled throughout the dynamics as a function of $|u_i^{\mathrm{R}} + u_i^{\mathrm{L}} - 2v_i|$.

We can now state the Proposition showing the recovery in finite time. The proof resembles the sketch of Section 4.2 with additional technical details since we need to control the distance between neurons, and the fact that $\eta > 0$ makes the dynamics more delicate to describe.

**Proposition 3.** *Let $f^* \in \mathcal{F}_{n, \Delta v, \Delta f, M}$. Assume that the initialization $u(0)$ is $D$-good with $D = 2^{13/2}(m+1)^{1/2} M \eta^{1/2} (\Delta f)^{-1}$. Then the maximal solution of (8) taking values in $\mathcal{U}$ is defined at least on $[0, \mathcal{T}]$ for $\mathcal{T} = 6/(\Delta f)^2$, and at the end of this time interval, there is a neuron at distance less than $\eta$ from each discontinuity of $f^*$.*

This Proposition is the main building block to show recovery in the next Theorem, along with some high-probability bounds to ensure that an i.i.d. uniform initialization is $D$-good.

**Theorem 2.** *Let $\xi, \delta > 0$, and $f^*$ a piecewise constant function from $\mathcal{F}_{n, \Delta v, \Delta f, M}$. Assume that the neural network has $m$ neurons with*

$$m \geqslant \frac{6}{\Delta v} \Big( 4 + \log n + \log \frac{1}{\delta} \Big) \,.$$

*Assume that, at initialization, the positions $u_1, \ldots, u_m$ of the neurons are i.i.d. uniformly distributed on $[0, 1]$. Then there exists $Q$ depending on $\xi, \delta, m, \Delta f, M$ such that, if*

$$\eta \leqslant Q \,,$$

*then, with probability at least $1 - \delta$, the maximal solution to the two-timescale limit (8) is defined at least until $\mathcal{T} = \frac{6}{(\Delta f)^2}$, and*

$$\int_0^1 |f^*(x) - f(x; a^*(u(\mathcal{T})), u(\mathcal{T}))|^2 \mathrm{d}x \leqslant \xi \,.$$

*Furthermore, we have $Q = \dfrac{C}{M^2} \min \Big( \dfrac{\delta^2 (\Delta f)^2}{(m+1)^5}, \dfrac{\xi}{n} \Big)$ for some universal constant $C > 0$.*

### 5.2 From the two-timescale limit to the two-timescale regime

We now briefly explain how the proof for the two-timescale limit can be adapted for the gradient flow problem in the two-timescale regime (1), that is with a small but non-vanishing $\varepsilon$. First note that the existence and uniqueness of the maximal solution to the dynamics (1) follow from the local Lipschitz-continuity of $\nabla_a L$ and $\nabla_u L$ with respect to both their variables.

The heuristics of Section 4.1 indicate that, for $\varepsilon$ small enough, at any time $t$, the weights $a(t)$ are close to $a^*(u(t))$, the global minimizer of $L(\cdot, u(t))$. The next Proposition formalizes this intuition.

**Proposition 4.** *Assume that $a(0) = 0$ and that, for all $s \in [0, t]$, there at least $2$ positions $u_j(s)$ in each interval $[v_i, v_{i+1}]$ and $\Delta(u(s)) \geqslant D/2$ for some $D \geqslant 32\eta$. Finally, assume that $\varepsilon \leqslant 2^{-16} D^2 M^{-2} (m+1)^{-5/2}$. Then*

$$\|a(t) - a^*(u(t))\| \leqslant 3M\sqrt{m+1} \exp^{-\frac{D}{16}t} + \frac{2^{17} M^3 (m+1)^3}{D^2} \varepsilon \,.$$

The crucial condition in the Proposition is $\Delta(u(s)) \geqslant D/2$; it is useful to control the conditioning of the quadratic form $L(\cdot, u(s))$. The Proposition shows that $\|a(t) - a^*(u(t))\|$ is upper bounded by the sum of two terms; the first term is a consequence to the initial gap between $a(0)$ and $a^*(u(0))$ and decays exponentially quickly. The second term is negligible in the regime $\varepsilon \ll 1$.

Armed with this Proposition, we show that the two-timescale regime has the same behavior as the two-timescale limit and thereby prove Theorem 1.

## 6 Numerical experiments and discussion

**Numerical illustration in the setting of Section 2.** We first compare the dynamics of the gradient flow in the two-timescale limit presented in Section 4.2 with the dynamics of SGD. To simulate the SGD dynamics, we assume that we have access to noisy observations of the value of $f^* \in \mathcal{F}_{n, \Delta v, \Delta f, M}$: let $(X_p, Y_p)_{p \geqslant 1}$ be i.i.d. random variables such that $X_p$ is uniformly distributed on $[0, 1]$, and $Y_p = f^*(X_p) + N_p$ where $N_p$ is additive noise. The (one-pass) SGD updates are then given by

$$\begin{aligned}
a_{p+1} &= a_p - h\nabla_a \ell(X_{p+1}, Y_{p+1}; a_p, u_p) \,, \\
u_{p+1} &= u_p - \varepsilon h \nabla_u \ell(X_{p+1}, Y_{p+1}; a_p, u_p) \,,
\end{aligned} \tag{9}$$

with $\ell(X, Y; a, u) = \frac{1}{2}(Y - f(X; a, u))^2$. The experimental settings, as well as additional results, are given in the Appendix.

Remarkably, the dynamics of SGD in the two-timescale regime with $\eta$ small match closely the gradient flow in the two-timescale limit with $\eta = 0$, as illustrated in Figure 3. This validates the use

of the gradient flow to understand the training dynamics with SGD. Both dynamics are close until the two-timescale limit achieves perfect recovery of the target function, at which point the SGD stabilizes to a small non-zero error. The fact that SGD does not achieve perfect recovery is not surprising, since SGD is performed with $\eta > 0$ and $f^*$ is not in the span of $\{1, \sigma_\eta(x - u_1), \dots, \sigma_\eta(x - u_m)\}$ for any $u_1, \dots, u_m$ and for $\eta > 0$. On the contrary, we simulated the dynamics of gradient flow for $\eta = 0$, as presented in Section 4.2, enabling perfect recovery in that case.

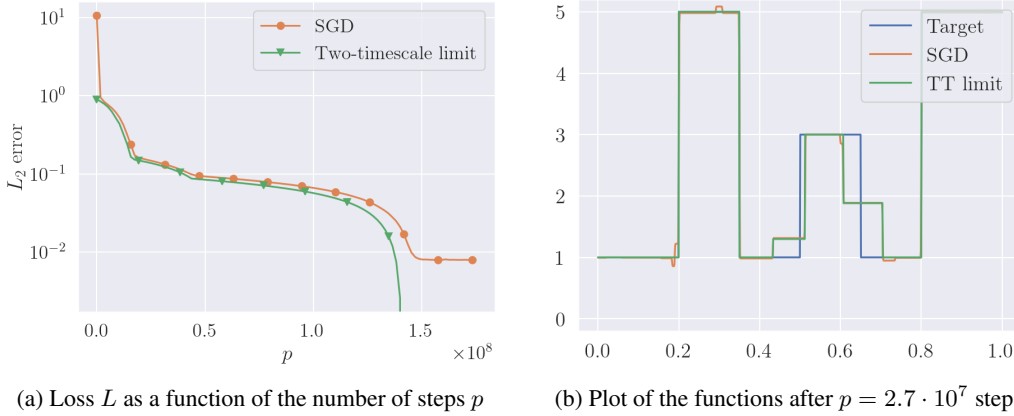

(a) Loss $L$ as a function of the number of steps $p$

(b) Plot of the functions after $p = 2.7 \cdot 10^7$ steps

Figure 3: Comparison between the SGD (9) with $\eta = 4 \cdot 10^{-3}$ in the two-timescale regime ($\varepsilon = 2 \cdot 10^{-5}$) and the gradient flow in the two-timescale limit (5) with $\eta = 0$. In the left-hand plot, to align the SGD and the two-timescale limit, we take $\tau = \varepsilon h p$. In the right-hand plot, the target function is in blue, the gradient flow in the two-timescale limit is in green, and the SGD is in orange.

Next, we compare the SGD dynamics in the two-timescale regime ($\varepsilon \ll 1$) and outside of this regime ($\varepsilon \approx 1$). In Figure 4, we see that the network trained by SGD (in orange) in the two-timescale regime $\varepsilon = 2 \cdot 10^{-5}$, achieves near-perfect recovery. If we change $\varepsilon$ to 1, while keeping all other parameters equal, the algorithm fails to recover the target function (Figure 5). This shows that, in our setting with a moderate number of neurons $m$, recovery can fail away from the two-timescale regime. It could seem that we are favouring the two-timescale regime by running it for more steps. In fact, it is not the case since both regimes are run until convergence. We refer to Appendix C for details.

Note that the dynamics of the neurons in Figures 4 and 5 are different. In the two-timescale regime, only the neurons closest to a discontinuity move significantly, while the others do not. These dynamics correspond to the sketch of Section 4. Interestingly, it means that in this regime, the neural network learns a sparse representation of the target function, meaning that only $n$ out of the $m$ neurons are active after training. On the contrary, when $\varepsilon = 1$, all neurons move to align with discontinuities of the target function, thus the learned representation is not sparse. Furthermore, since the number of neurons is moderate, one of the discontinuities is left without any neuron.

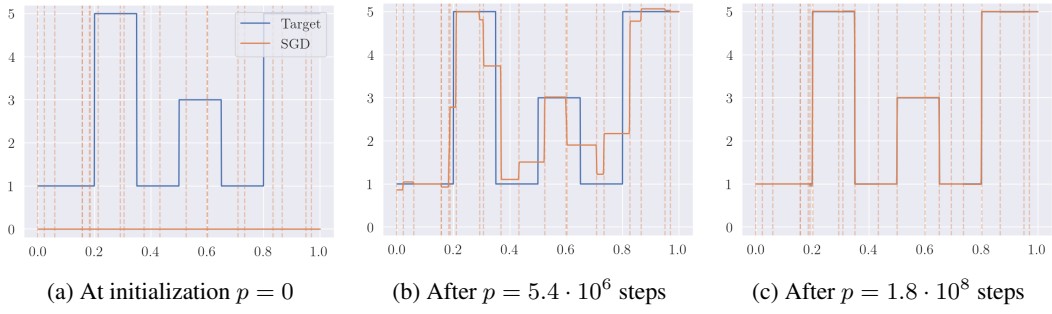

(a) At initialization $p = 0$

(b) After $p = 5.4 \cdot 10^6$ steps

(c) After $p = 1.8 \cdot 10^8$ steps

Figure 4: Simulation in the two-timescale regime ($\varepsilon = 2 \cdot 10^{-5}$). The target function is in blue and the SGD (9) is in orange with $\eta = 4 \cdot 10^{-3}$, $h = 10^{-5}$. The positions $u_1, \dots, u_m$ of the neurons are indicated with vertical dotted lines. In a first short phase, only the weights $a_1, \dots, a_m$ of the neurons evolve to match as best as possible the target function (second plot). Then, in a longer phase, the neuron closest to each target discontinuity moves towards it (third plot). Recovery is achieved.

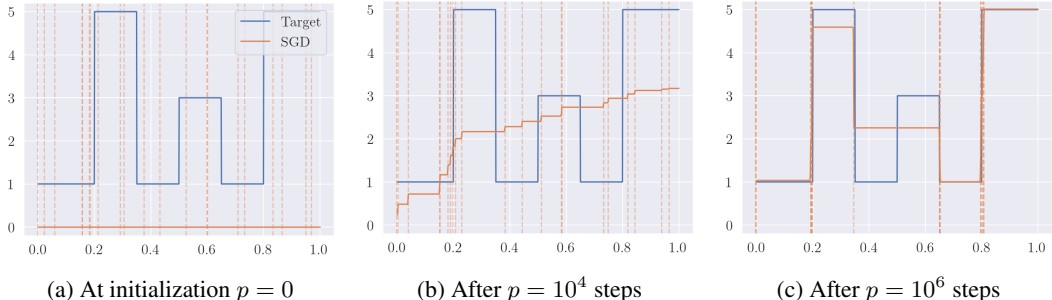

(a) At initialization $p = 0$     (b) After $p = 10^4$ steps     (c) After $p = 10^6$ steps

Figure 5: Simulation outside of the two-timescale regime ($\varepsilon = 1$). The target function is in blue and the SGD (9) is in orange with $\eta = 4 \cdot 10^{-3}$, $h = 10^{-5}$. The positions $u_1, \ldots, u_m$ of the neurons are indicated with vertical dotted lines. The dynamics create a zone with no neuron, hindering recovery.

**Discussion.** The two-timescale regime decouples the dynamics of the two layers of the neural network. As a consequence, it is a useful theoretical tool to simplify the evolution equations. In this paper showcasing the approach, the two-timescale regime enables to show the alignment of the neurons with the discontinuities of the target function in the piecewise constant 1D case, and thus to prove recovery. A full general understanding of the impact of the two-timescale regime is an open question, which is left for future work. We provide in the following some practical evidence of the applicability of this regime to other settings (higher-dimensional problems, ReLU networks, finite sample size). Additional technical difficulties significantly complicate the proof in these settings, but we believe that there is no fundamental reason that our mathematical approach should not apply.

**Higher dimensions.** We consider piecewise constant functions on $\mathbb{R}^d$ with pieces that are cuboids aligned with the axes of the space (see Figure 6). Neural networks are of the form $f(x; a, u) = a_0 + \sum_{j=1}^{m} \sum_{k=1}^{d} a_{jk} \sigma_\eta(x_k - u_{jk})$, where the $j$-th neuron has $d$-dimensional position $(u_{jk})_{1 \leqslant k \leqslant d}$ and weight $(a_{jk})_{1 \leqslant k \leqslant d}$. The results are similar to the 1D case: convergence to a global minimum is obtained in the two-timescale regime but not in the standard regime.

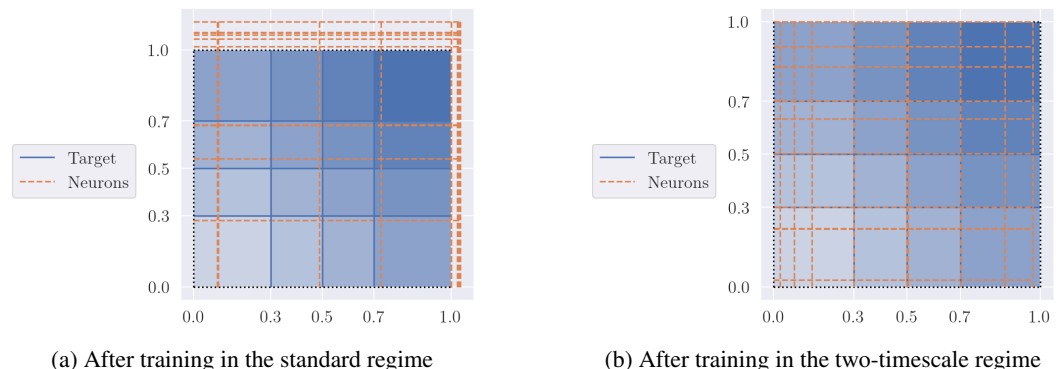

(a) After training in the standard regime     (b) After training in the two-timescale regime

Figure 6: 2D experiment with a piecewise-constant target (each shade of blue depicts a constant piece). The orange lines show the positions of the neurons after training by SGD. In the standard regime, a discontinuity of the target at $x = 0.3$ is not covered by a neuron. In the two-timescale regime, all the target discontinuities are covered. See Appendix C for more results with $d = 2, 10$.

**ReLU networks.** Appendix C reports the case of using ReLU activations to approximate piecewise-affine targets. A similar conclusion holds.

**Finite sample size.** We believe that it should be possible to generalize our results to finite sample sizes (say, for single-pass SGD), following a perturbation analysis similar to the one of Section 5.2, with additional terms due to random sampling. Numerically, Figure 3 shows that SGD indeed closely follows the behavior of the two-timescale limit. Some ideas about the proof are provided in Appendix D.2.

## Acknowledgments and Disclosure of Funding

The authors thank Emmanuel Abbé for many discussions and suggestions on this project, and Quentin Berthet for pointing them towards the envelope theorem. The authors are grateful towards the Simons Institute for the Theory of Computing where this work was initiated. P.M. is supported by a grant from Région Île-de-France, by a Google PhD Fellowship and by Mines Paris - PSL.

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

# Appendix

**Organization of the Appendix**   In Section A, we introduce additional notations that will be used throughout the Appendix, then proceed to prove useful technical lemmas. We proceed in Section B to prove the results presented in the main text. Section C contains details about our experimental settings as well as some additional simulations. Section D gives a couple of additional insights about our results.

## A   Additional notations and technical lemmas

For a vector $a$, we denote $\|a\|$ its $\ell^2$-norm, $\|a\|_1$ its $\ell^1$-norm and $\|a\|_\infty$ its $\ell^\infty$-norm. For matrices $H$, $\|H\|$ denotes the operator norm associated to the $\ell^2$ norm and $\|H\|_F$ denotes the Frobenius norm. For real-valued functions $f$, $\|f\|_\infty$ denotes the supremum norm.

In all of the Appendix, we denote $u_0 = -\eta/2$ and $u_{m+1} = 1 + \eta/2$. Note that $\sigma_\eta(x - u_0) = 1$ for all $x \in [0, 1]$, meaning that $\sigma_\eta(\cdot - u_0)$ corresponds to the bias term. This notation allows to treat the bias term in a unified fashion with respect to the other terms of $f(x; a, u)$. Since $u_i \in (0, 1)$ for $i \in \{1, \ldots, m\}$, we assume in the following w.l.o.g. that the $(u_i)_{0 \leqslant i \leqslant m+1}$ are ordered in increasing order. Note that we prove in the following that the $(u_i)_{1 \leqslant i \leqslant m}$ do not cross during the dynamics, so they remain ordered throughout the dynamics.

The proofs involve comparisons of some quantities when $\eta > 0$ and when $\eta = 0$. To avoid confusion, we make explicit the dependency of $L$ on $\eta \geqslant 0$, i.e., we let $L_\eta(a, u)$ in place of $L(a, u)$ of the main paper, and similarly, when the $\arg\min$ is well-defined and unique,

$$a_\eta^*(u) = \underset{a \in \mathbb{R}^{m+1}}{\arg\min} \, L_\eta(a, u) \,.$$

in place of $a^*(u)$. Similarly, we now make explicit the dependence of $f$ on $\eta \geqslant 0$, i.e., we denote

$$f_\eta(x; a, u) = a_0 + \sum_{j=1}^m a_j \sigma_\eta(x - u_j) = \sum_{j=0}^m a_j \sigma_\eta(x - u_j) \,.$$

The Hessian of the quadratic function $L_\eta(\cdot, u)$ is denoted $H_\eta(u) \in \mathbb{R}^{(m+1) \times (m+1)}$ (in place of $H(u)$), and satisties that, for $i, j \in \{0, \ldots, m\}$,

$$H_{\eta,ij}(u) = \int_0^1 \sigma_\eta(x - u_i) \sigma_\eta(x - u_j) \mathrm{d}x \,.$$

Also let, for $\eta \geqslant 0$ and $u \in \mathbb{R}^m$, $b_\eta(u) \in \mathbb{R}^{m+1}$ such that, for $j \in \{0, \ldots, m\}$,

$$b_{\eta,j}(u) = \int_0^1 f^*(x) \sigma_\eta(x - u_j) \mathrm{d}x \,.$$

Finally, we let $\mathcal{U}_\eta$ in place of $\mathcal{U}$ in the paper.

With these notations, we have, for $\eta \geqslant 0$ and $a, u \in \mathbb{R}^m$,

$$\frac{\partial L_\eta}{\partial u_j}(a, u) = \int_0^1 \frac{\partial f_\eta(x; a, u)}{\partial u_j} (f_\eta(x; a, u) - f^*(x)) \, \mathrm{d}x$$

$$= -a_j \int_0^1 \sigma_\eta'(x - u_j) \Big( \sum_{k=0}^m a_k \sigma_\eta(x - u_k) - f^*(x) \Big) \mathrm{d}x \,. \tag{10}$$

and

$$\frac{\partial L_\eta}{\partial a_j}(a, u) = \int_0^1 \frac{\partial f_\eta(x; a, u)}{\partial a_j} (f_\eta(x; a, u) - f^*(x)) \, \mathrm{d}x$$

$$= \int_0^1 \sigma_\eta(x - u_j) \Big( \sum_{k=0}^m a_k \sigma_\eta(x - u_k) - f^*(x) \Big) \mathrm{d}x$$

$$= H_{\eta,j}(u)^\top a - b_{\eta,j}(u) \,. \tag{11}$$

We now move on to a series to lemmas that will be helpful in the proofs of Appendix B.

**Lemma 1.** *For $\eta \geqslant 0$ and $u \in \mathbb{R}^m$, we have*

$$\|b_\eta(u) - b_0(u)\| \leqslant M\eta\sqrt{m+1} \quad \text{and} \quad \|b_\eta(u)\| \leqslant M\sqrt{m+1}.$$

*Proof.* For any $j \in \{0, \ldots, m\}$,

$$|b_{\eta,j}(u) - b_{0,j}(u)| = \left| \int_0^1 f^*(x)(\sigma_\eta(x - u_j) - \sigma_0(x - u_j))\mathrm{d}x \right|$$

$$\leqslant \|f^*\|_\infty \int_0^1 |\sigma_\eta(x - u_j) - \sigma_0(x - u_j)| \, \mathrm{d}x$$

$$\leqslant M\eta,$$

where in the last step we use that $\|f^*\|_\infty \leqslant M$ and that $\sigma_\eta(x) = 0$ for $x \leqslant -\eta/2$, $\sigma_\eta(x) \in [0,1]$ for $-\eta/2 < x < \eta/2$ and $\sigma_\eta(x) = 1$ for $x \geqslant \eta/2$.

Similarly,

$$|b_{\eta,j}(u)| = \left| \int_0^1 f^*(x)\sigma_\eta(x - u_j)\mathrm{d}x \right| \leqslant \|f^*\|_\infty \leqslant M.$$

$\square$

**Lemma 2.** *For $\eta \geqslant 0$ and $u \in \mathcal{U}_\eta$, $H_\eta(u) = H_0(u) + D_\eta$, where $D_\eta$ is a diagonal matrix whose elements are independent of $u$ and bounded in absolute value by $\eta/2$.*

*Proof.* Let $i, j \in \{0, \ldots, m\}$, and denote $c = \max(u_i, u_j, 0)$. Then

$$H_{0,ij}(u) = \int_0^1 \sigma_0(x - u_i)\sigma_0(x - u_j)dx = 1 - c.$$

If $i = j = 0$, $\max(u_i, u_j) = -\eta/2$, and $H_{\eta,ij}(u) = 1 = H_{0,ij}(u)$. If $i = j \neq 0$,

$$H_{\eta,ij}(u) = \int_0^1 \sigma_\eta(x - c)^2 dx$$

$$= 1 - c - \frac{\eta}{2} + \int_{c-\eta/2}^{c+\eta/2} \sigma_\eta(x - c)^2 dx$$

$$= H_{0,ij}(u) - \frac{\eta}{2} + \eta \int_{-1/2}^{1/2} \sigma^2.$$

Note that the last integral is non-negative and less than 1, hence $|H_{\eta,ij}(u) - H_{0,ij}(u)| \leqslant \eta/2$. Finally, if $i \neq j$, since $|u_i - u_j| > \eta$,

$$H_{\eta,ij}(u) = \int_0^1 \sigma_\eta(x - u_i)\sigma_\eta(x - u_j)dx = \int_0^1 \sigma_\eta(x - \max(u_i, u_j))dx.$$

Furthermore, $0 < \max(u_i, u_j) < 1 - \frac{\eta}{2}$, thus

$$H_{\eta,ij}(u) = \int_0^1 \sigma_\eta(x - c)dx = 1 - c - \frac{\eta}{2} + \int_{c-\eta/2}^{c+\eta/2} \sigma_\eta(x - c)dx = 1 - c,$$

where the last equality comes from the oddness of $\sigma - 1/2$.

$\square$

**Lemma 3.** *For $\eta > 0$, let $a_\eta^* : u \in \mathcal{U}_\eta \mapsto a_\eta^*(u)$. Then $a_\eta^*$ is differentiable and for any $u \in \mathcal{U}_\eta$,*

$$\left\| \frac{\partial a_\eta^*(u)}{\partial u} \right\| \leqslant \frac{8}{\Delta(u)} \Big( 2(m+1)\|a_\eta^*(u)\| + M \Big).$$

*Proof.* By Proposition 1 (whose proof does not rely on this lemma), for $u \in \mathcal{U}_\eta$, $L_\eta(\cdot, u)$ has a unique minimizer $a_\eta^*(u)$, which is equal to $H_\eta(u)^{-1}b_\eta(u)$ by (11). Furthermore, $H_\eta$ and $b_\eta$ are differentiable with respect to $u$, hence $a_\eta^*$ is also differentiable with respect to $u$, and we have

$$\frac{\partial a_\eta^*(u)}{\partial u_k} = -H_\eta(u)^{-1}\frac{\partial H_\eta}{\partial u_k}(u)a_\eta^*(u) + H_\eta(u)^{-1}\frac{\partial b_\eta}{\partial u_k}(u).$$

Denote $w_k(u) := \frac{\partial H_\eta}{\partial u_k}(u)a_\eta^*(u)$ and $W(u)$ the $(m+1) \times (m+1)$ matrix formed by stacking column-wise the vectors $(w_k(u))_{0 \leqslant k \leqslant m}$. Then

$$\frac{\partial a_\eta^*(u)}{\partial u} = -H_\eta(u)^{-1}W(u) + H_\eta(u)^{-1}\frac{\partial b_\eta}{\partial u}(u).$$

We now estimate the Frobenius norm of the matrix $W(u)$. By Lemma 2, for $u \in \mathcal{U}_\eta$, $H_\eta(u) = H_0(u) + D_\eta$. Take $i,j \in \{0, \ldots, m\}$, then

$$H_{\eta,ij}(u) = H_{0,ij}(u) + D_{\eta,ij} = \int_0^1 \sigma_0(x-u_i)\sigma_0(x-u_j)dx + D_{\eta,ij} = 1 - \max(u_i, u_j, 0) + D_{\eta,ij}.$$

Hence $\frac{\partial H_{\eta,ij}}{\partial u_k} = 0$ if $i, j \neq k$. Further, if $i = k$ and $j \neq k$,

$$\left|\frac{\partial H_{\eta,ij}}{\partial u_k}(u)\right| = \left|\frac{\partial}{\partial u_i}(1 - \max(u_i, u_j))\right| \leqslant 1.$$

Of course, the bound $|\frac{\partial H_{\eta,ij}}{\partial u_k}(u)| \leqslant 1$ also holds when $j = k$ and $i \neq j$. Finally, a similar bound shows that $|\frac{\partial H_{\eta,ij}}{\partial u_k}(u)| \leqslant 2$ when $i = j = k$.

As a consequence, for $k, i \in \{0, \ldots, m\}$,

$$|w_{k,i}(u)| \leqslant \sum_{j=0}^m \left|\frac{\partial H_{\eta,ij}}{\partial u_k}(u)\right| |a_{\eta,j}^*(u)| \leqslant \begin{cases} |a_{\eta,k}^*(u)| & \text{if } i \neq k, \\ |a_{\eta,k}^*(u)| + \|a_\eta^*(u)\|_1 & \text{if } i = k. \end{cases}$$

Thus

$$\|W(u)\|_F = \left(\sum_{i=0}^m \sum_{k=0}^m |w_{k,i}(u)|^2\right)^{1/2} \leqslant \left(\sum_{i=0}^m \left(\sum_{k=0}^m |w_{k,i}(u)|\right)^2\right)^{1/2}$$

$$\leqslant \left(\sum_{i=0}^m \left(2\|a_\eta^*(u)\|_1\right)^2\right)^{1/2} = 2\sqrt{m+1}\|a_\eta^*(u)\|_1.$$

With a reasoning similar to the above, note that $\frac{\partial b_\eta}{\partial u}(u)$ is a diagonal matrix with diagonal entries in $[-M, M]$. Finally, putting these elements together, using Proposition 1 and that $\|W(u)\| \leqslant \|W(u)\|_F$, we obtain

$$\left\|\frac{\partial a_\eta^*(u)}{\partial u}\right\| \leqslant \|H_\eta(u)^{-1}\|\|W(u)\|_F + \|H_\eta(u)^{-1}\|\left\|\frac{\partial b_\eta(u)}{\partial u}\right\| \leqslant \frac{8}{\Delta(u)}\left(2\sqrt{m+1}\|a_\eta^*(u)\|_1 + M\right).$$

$\square$

The following lemma gives exact formulae for the derivative of the loss $L_\eta$ with respect to the positions of the neurons, evaluated for $a = a_0^*(u)$, that is the best piecewise constant approximation of $f^*$ with subdivision $\{u_1, \ldots, u_m\}$. Note that the formulae are the same as in Section 4.2, but the derivation is slightly more intricate since we consider here the loss $L_\eta$ and not $L_0$.

**Lemma 4.** *Take $\eta > 0$ and $u \in \mathcal{U}_\eta$ such that there are at least two neurons on each piece $[v_i, v_{i+1}]$ of $f^*$. Then, if $u_j$ does not flank a discontinuity of $f^*$,*

$$\frac{\partial L_\eta}{\partial u_j}(a_0^*(u), u) = 0.$$

*Furthermore, for a discontinuity $v_i$, denote $u_i^L$ is the closest neuron to its left and $u_i^R$ the closest neuron to its right. If $v_i - u_i^L \geqslant \frac{\eta}{2}$ and $u_i^R - v_i \geqslant \frac{\eta}{2}$, then*

$$\frac{\partial L_\eta}{\partial u_i^L}(a_0^*(u), u) = -\frac{1}{2}\frac{(u_i^R - v_i)^2}{(u_i^R - u_i^L)^2}(f_i^* - f_{i-1}^*)^2,$$

$$\frac{\partial L_\eta}{\partial u_i^R}(a_0^*(u), u) = \frac{1}{2}\frac{(v_i - u_i^L)^2}{(u_i^R - u_i^L)^2}(f_i^* - f_{i-1}^*)^2.$$

*Proof.* In this proof, let us denote for simplicity $a = a_0^*(u)$. At the condition that there is at least two neurons on each piece of $f^*$, Section 4.2 gives the optimal approximation $f_0(x; a, u)$ of $f^*$ that is piecewise constant with respect to the subdivision $\{u_1, \ldots, u_m\}$. As a consequence, we easily get the value of $a$. Namely, if $u_j$ does not flank a discontinuity of $f^*$, the value of $f_0(x; a, u)$ is locally constant around $u_j$, thus $a_j = 0$. Plugging into (10), we obtain

$$\frac{\partial L_\eta}{\partial u_j}(a, u) = 0.$$

Further, for a discontinuity $v_i$, denote respectively $a_i^{\mathrm{L}}$ and $a_i^{\mathrm{R}}$ the coefficients associated to $u_i^{\mathrm{L}}$ and $u_i^{\mathrm{R}}$. At $u_i^{\mathrm{L}}$, the value of $f_0(x; a, u)$ jumps from $f_{i-1}^*$ to $\frac{v_i - u_i^{\mathrm{L}}}{u_i^{\mathrm{R}} - u_i^{\mathrm{L}}} f_{i-1}^* + \frac{u_i^{\mathrm{R}} - v_i}{u_i^{\mathrm{R}} - u_i^{\mathrm{L}}} f_i^*$, thus

$$a_i^{\mathrm{L}} = \frac{v_i - u_i^{\mathrm{L}}}{u_i^{\mathrm{R}} - u_i^{\mathrm{L}}} f_{i-1}^* + \frac{u_i^{\mathrm{R}} - v_i}{u_i^{\mathrm{R}} - u_i^{\mathrm{L}}} f_i^* - f_{i-1}^* = \frac{u_i^{\mathrm{R}} - v_i}{u_i^{\mathrm{R}} - u_i^{\mathrm{L}}} (f_i^* - f_{i-1}^*).$$

Similarly, we have

$$a_i^{\mathrm{R}} = \frac{v_i - u_i^{\mathrm{L}}}{u_i^{\mathrm{R}} - u_i^{\mathrm{L}}} (f_i^* - f_{i-1}^*).$$

We now compute, using (10),

$$\frac{\partial L_\eta}{\partial u_i^{\mathrm{L}}}(a, u) = -a_i^{\mathrm{L}} \int_0^1 \sigma_\eta'(x - u_i^{\mathrm{L}}) (f_\eta(x; a, u) - f^*(x)) \, \mathrm{d}x$$

$$= -a_i^{\mathrm{L}} \int_{u_i^{\mathrm{L}} - \eta/2}^{u_i^{\mathrm{L}} + \eta/2} \sigma_\eta'(x - u_i^{\mathrm{L}}) (f_\eta(x; a, u) - f^*(x)) \, \mathrm{d}x.$$

Using that $\Delta(u) > 2\eta$ and that there are at least two neurons on each piece of $f^*$, we have that $u_i^{\mathrm{L}} - v_{i-1} \geqslant 2\eta$. Since, in addition, by assumption, $v_i - u_i^{\mathrm{L}} \geqslant \frac{\eta}{2}$, we get that for $x \in \left[u_i^{\mathrm{L}} - \frac{\eta}{2}, u_i^{\mathrm{L}} + \frac{\eta}{2}\right]$, $f^*(x) = f_{i-1}^*$. Moreover, using again $\Delta(u) \geqslant 2\eta$ that $\sigma_\eta$ is equal to $\sigma_0$ on $(-\infty, -\eta/2]$ and $[\eta/2, \infty)$, we have for $x \in \left[u_i^{\mathrm{L}} - \frac{\eta}{2}, u_i^{\mathrm{L}} + \frac{\eta}{2}\right]$,

$$f_\eta(x; a, u) = \sum_{k=0}^m a_k \sigma_\eta(x - u_k) = f_0\left(u_i^{\mathrm{L}} - \frac{\eta}{2}; a, u\right) + a_i^{\mathrm{L}} \sigma_\eta(x - u_i^{\mathrm{L}}) = f_{i-1}^* + a_i^{\mathrm{L}} \sigma_\eta(x - u_i^{\mathrm{L}}).$$

Thus we obtain

$$\frac{\partial L_\eta}{\partial u_i^{\mathrm{L}}}(a, u) = -a_i^{\mathrm{L}} \int_{u_i^{\mathrm{L}} - \eta/2}^{u_i^{\mathrm{L}} + \eta/2} \sigma_\eta'(x - u_i^{\mathrm{L}}) a_i^{\mathrm{L}} \sigma_\eta(x - u_i^{\mathrm{L}}) \mathrm{d}x$$

$$= -\frac{(a_i^{\mathrm{L}})^2}{2} \left(\sigma_\eta\left(\frac{\eta}{2}\right)^2 - \sigma_\eta\left(-\frac{\eta}{2}\right)^2\right)$$

$$= -\frac{(a_i^{\mathrm{L}})^2}{2}$$

$$= -\frac{1}{2} \frac{(u_i^{\mathrm{R}} - v_i)^2}{(u_i^{\mathrm{R}} - u_i^{\mathrm{L}})^2} (f_i^* - f_{i-1}^*)^2.$$

The computation of $\frac{\partial L_\eta}{\partial u_i^{\mathrm{R}}}(a, u)$ is similar.

$\square$

**Lemma 5.** *Consider $\eta \geqslant 0$ and $u \in \mathcal{U}_\eta$ such that there are at least two neurons on each piece $[v_i, v_{i+1}]$ of $f^*$. Then, for all $x \in [0, 1]$, $|f_\eta(x; a_0^*(u), u)| \leqslant M$.*

*Proof.* In the case where $\eta = 0$, the result easily follows from the expressions for $f_0(x; a_0^*(u), u)$ provided in Section 4.2. We now assume $\eta > 0$.

Denote $A_k^*(u) = \sum_{j=0}^{k} a_{0,j}^*(u)$ (with the convention $A_{-1}^*(u) = 0$). Recall the convention $u_0 = -\eta/2$. We compute

$$
\begin{aligned}
f_\eta(x; a_0^*(u), u) &= \sum_{k=0}^{m} a_{0,k}^*(u)\sigma_\eta(x - u_k) \\
&= \sum_{k=0}^{m} \left( A_k^*(u) - A_{k-1}^*(u) \right) \sigma_\eta(x - u_k) \\
&= \sum_{k=0}^{m-1} A_k^*(u) \left( \sigma_\eta(x - u_k) - \sigma_\eta(x - u_{k+1}) \right) + A_m^*(u)\sigma_\eta(x - u_m)
\end{aligned}
$$

Note that $A_k^*(u) = \lim_{x \to u_k+} f_0(x; a_0^*(u), u)$, and thus, from the case $\eta = 0$, we have $|A_k^*(u)| \leqslant M$. Moreover, $\sigma_\eta$ is increasing and the $u_k$ are in increasing order. We thus get

$$
\begin{aligned}
|f_\eta(x; a_0^*(u), u)| &\leqslant M \Big( \sum_{k=0}^{m-1} (\sigma_\eta(x - u_k) - \sigma_\eta(x - u_{k+1})) + \sigma_\eta(x - u_m) \Big) \\
&= M\sigma_\eta(x - u_0) \leqslant M \, .
\end{aligned}
$$

$\square$

**Lemma 6.** *Consider $\eta > 0$ and $u \in \mathcal{U}_\eta$ such that there are at least two neurons on each piece $[v_i, v_{i+1}]$ of $f^*$. Then, for $j \in \{0, \dots, m\}$,*

$$
|a_{0,j}^*(u)| \leqslant 2M
$$

*and, for any $a \in \mathbb{R}^{m+1}$,*

$$
\left| \frac{\partial L_\eta}{\partial u_j}(a, u) - \frac{\partial L_\eta}{\partial u_j}(a_0^*(u), u) \right| \leqslant 2M(\sqrt{m+1}+1)\|a - a_0^*(u)\| + \sqrt{m+1}\|a - a_0^*(u)\|^2 \, .
$$

*Proof.* The first statement of the Lemma comes from the explicit formulae for $a_0^*(u)$ given in the proof of Lemma 4, namely each $a_{0,j}^*(u)$ is either zero or less in magnitude than the gap between two pieces of $f^*$ that is less than $2M$.

By (10), we have

$$
\begin{aligned}
&\left| \frac{\partial L_\eta}{\partial u_j}(a, u) - \frac{\partial L_\eta}{\partial u_j}(a_0^*(u), u) \right| \\
&\qquad = \left| a_j \int_0^1 \sigma_\eta'(x - u_j)\Big( f_\eta(x; a, u) - f^*(x) \Big) \mathrm{d}x \right. \\
&\qquad\qquad \left. - a_{0,j}^*(u) \int_0^1 \sigma_\eta'(x - u_j)\Big( f_\eta(x; a_0^*(u), u) - f^*(x) \Big) \mathrm{d}x \right| \\
&\qquad \leqslant |a_j - a_{0,j}^*(u)| \int_0^1 \sigma_\eta'(x - u_j) |f_\eta(x; a_0^*(u), u) - f^*(x)| \, \mathrm{d}x \\
&\qquad\qquad + |a_j| \int_0^1 \sigma_\eta'(x - u_j) |f_\eta(x; a, u) - f_\eta(x; a_0^*(u), u)| \, \mathrm{d}x \, .
\end{aligned}
$$

We bound the two terms separately. For the first term, we use Lemma 5.

$$
\begin{aligned}
\int_0^1 \sigma_\eta'(x - u_j) |f_\eta(x; a_0^*(u), u) - f^*(x)| &\leqslant \int_0^1 \sigma_\eta'(x - u_j) \left( |f_\eta(x; a_0^*(u), u)| + |f^*(x)| \right) \\
&\leqslant 2M \int_0^1 \sigma_\eta'(x - u_j)\mathrm{d}x \leqslant 2M \, .
\end{aligned}
$$

We now continue with the second term.

$$
|f_\eta(x; a, u) - f_\eta(x; a_0^*(u), u)| = \Big| \sum_{k=0}^{m} (a_k - a_{0,k}^*(u))\sigma_\eta(x - u_k) \Big| \leqslant \|a - a_0^*(u)\|_1 \, ,
$$

and thus
$$\int_0^1 \sigma_\eta'(x - u_j) \, |f_\eta(x; a, u) - f_\eta(x; a_0^*(u), u)| \, \mathrm{d}x \leqslant \|a - a_0^*(u)\|_1 \int_0^1 \sigma_\eta'(x - u_j)\mathrm{d}x$$
$$\leqslant \|a - a_0^*(u)\|_1 \,.$$

Returning to our initial upper bound, we obtain, using the first statement of the Lemma,
$$\left| \frac{\partial L_\eta}{\partial u_j}(a, u) - \frac{\partial L_\eta}{\partial u_j}(a_0^*(u), u) \right| \leqslant 2M\|a - a_0^*(u)\| + (|a_{0,j}^*(u)| + |a_j - a_{0,j}^*(u)|)\|a - a_0^*(u)\|_1$$
$$\leqslant 2M\|a - a_0^*(u)\| + (2M + \|a - a_0^*(u)\|)\sqrt{m+1}\|a - a_0^*(u)\|$$
$$= 2M(\sqrt{m+1} + 1)\|a - a_0^*(u)\| + \sqrt{m+1}\|a - a_0^*(u)\|^2 \,.$$
$\square$

**Lemma 7.** *For $\eta \geqslant 0$ and $u \in \mathcal{U}_\eta$,*
$$\|a_\eta^*(u) - a_0^*(u)\| \leqslant \frac{16M\sqrt{m+1}\eta}{\Delta(u)} \,.$$

*Proof.* By (11),
$$H_\eta(u)a_\eta^*(u) = b_\eta(u)$$
and by (11) and by Lemma 2,
$$H_\eta(u)a_0^*(u) = H_0(u)a_0^*(u) + D_\eta a_0^*(u) = b_0(u) + D_\eta a_0^*(u).$$
According to Proposition 1 (whose proof does not rely on this lemma), $H_\eta(u)$ is invertible with $\|H_\eta(u)^{-1}\| \leqslant 8/\Delta(u)$. We thus have
$$\|a_\eta^*(u) - a_0^*(u)\| = \|H_\eta(u)^{-1}(H_\eta(u)a_\eta^*(u) - H_\eta(u)a_0^*(u))\|$$
$$\leqslant \frac{8}{\Delta(u)}\|b_\eta(u) - b_0(u) - D_\eta a_0^*(u)\|$$
$$\leqslant \frac{8}{\Delta(u)}\big(\|b_\eta(u) - b_0(u)\| + \|D_\eta a_0^*(u)\|\big)$$
$$\leqslant \frac{8}{\Delta(u)}\Big(\|b_\eta(u) - b_0(u)\| + \frac{\eta}{2}\|a_0^*(u)\|\Big) \,.$$
The result then unfolds from Lemmas 1 and 6.
$\square$

**Lemma 8.** *Let $\eta > 0$, $u \in \mathbb{R}^m$ and $a, a' \in \mathbb{R}^{m+1}$. Then*
$$\|\nabla_u L_\eta(a, u)\| \leqslant \sqrt{m+1}\|a\|^2 + M\|a\| \,,$$
$$\|\nabla_a L_\eta(a, u)\| \leqslant \sqrt{m+1}(\|a\|\sqrt{m+1} + M) \,.$$
*As a consequence of the second inequality, by the fundamental theorem of calculus for line integrals,*
$$|L_\eta(a, u) - L_\eta(a', u)| \leqslant \sqrt{m+1}\left(\max(\|a\|, \|a'\|)\sqrt{m+1} + M\right)\|a - a'\| \,.$$

*Proof.* Recall that, for all $j \in \{1, \ldots, m\}$, and for all $a, u \in \mathbb{R}^m$,
$$\frac{\partial L_\eta}{\partial u_j}(a, u) = -a_j \int_0^1 \sigma_\eta'(x - u_j)\Big(\sum_{k=0}^m a_k\sigma_\eta(x - u_k) - f^*(x)\Big)\mathrm{d}x \,,$$
$$\frac{\partial L_\eta}{\partial a_j}(a, u) = \int_0^1 \sigma_\eta(x - u_j)\Big(\sum_{k=0}^m a_k\sigma_\eta(x - u_k) - f^*(x)\Big)\mathrm{d}x \,.$$
From the first equality, we have
$$\left| \frac{\partial L_\eta}{\partial u_j}(a, u) \right| \leqslant |a_j| \int_0^1 |\sigma_\eta'(x - u_j)|\Big(\sum_{k=1}^m |a_k|\sigma_\eta(x - u_k) + |f^*(x)|\Big)dx$$
$$\leqslant |a_j|(\|a\|_1 + M) \int_0^1 |\sigma_\eta'(x - u_j)|dx$$
$$\leqslant |a_j|(\|a\|_1 + M) \,.$$

As a consequence,

$$\|\nabla_u L_\eta(a, u)\| \leqslant \|a\|(\|a\|_1 + M) \leqslant \sqrt{m+1}\|a\|^2 + M\|a\|.$$

Similarly, from the second equality, we have

$$\left| \frac{\partial L_\eta}{\partial a_j}(a, u) \right| \leqslant \|a\|_1 + M.$$

As a consequence,

$$\|\nabla_a L_\eta(a, u)\| \leqslant \sqrt{m+1}(\|a\|_1 + M) = \sqrt{m+1}(\|a\|\sqrt{m+1} + M).$$

$\square$

**Lemma 9.** *Consider $\eta \geqslant 0$ and $u \in \mathcal{U}_\eta$ such that there is a neuron at distance less than $\eta$ from each discontinuity of $f^*$ and $3\eta \leqslant \Delta v$. Then*

$$\int_0^1 |f_\eta(x; a_\eta^*(u), u) - f^*(x)|^2 \mathrm{d}x \leqslant 6M^2 \eta n.$$

*Proof.* By definition of $a_\eta^*(u)$,

$$\int_0^1 |f_\eta(x; a_\eta^*(u), u) - f^*(x)|^2 \mathrm{d}x = \min_{a \in \mathbb{R}^{m+1}} \int_0^1 |f_\eta(x; a, u) - f^*(x)|^2 \mathrm{d}x.$$

Thus it is enough to exhibit some $a$ for which the latter integral is smaller than $6M^2 \eta n$ to conclude.

We construct such an $a$ as follows: set $a_0 = f^*(0)$, and for each discontinuity $v_i$, set the coefficient of a neuron at distance less than $\eta$ to the value $f_i^* - f_{i-1}^*$ and set all other neurons to zero. Note that the active neurons are distinct since $3\eta \leqslant \Delta v$.

Then the neural network is equal to the target function everywhere except on an interval of size $3\eta/2$ around each discontinuity, where they disagree (in infinite norm) by at most $2M$.

$\square$

**Lemma 10.** *Let $m$ be a positive integer and $u_1, \ldots, u_m$ be i.i.d. uniform random variables in $[0, 1]$. Assume that*

$$m \geqslant \frac{6}{\Delta v}\left(4 + \log n + \log \frac{1}{\delta}\right).$$

*Then, with probability at least $1 - \delta$, the vector $u$ is $D$-good with $D = \frac{\delta}{6(m+1)^2}$.*

*Proof.* We define the following events:

(a) $A$ is the event "there are at least 6 positions $u_j$ in each interval $[v_i, v_{i+1}]$ for $i \in \{0, \ldots, n-1\}$",

(b) $B$ is the event "$\Delta(u) \geqslant D$",

(c) for all $i \in \{1, \ldots, n-1\}$, $E_i$ is the event "there are at least one neuron on the left and on the right of $v_i$" and $C_i$ is the event "$E_i$ holds and $|u_i^{\mathrm{R}} + u_i^{\mathrm{L}} - 2v_i| \geqslant D$".

Note that by Definition 2, $u$ is $D$-good if and only if the event $A \cap B \cap (\bigcap_i C_i)$ holds. To show that this holds with high probability, we bound the probability of the complement

$$\left(A \cap B \cap \left(\bigcap_i C_i\right)\right)^c = A^c \cup B^c \cup \left(\bigcup_i C_i^c\right) = A^c \cup B^c \cup \left(\bigcup_i (C_i^c \cap A)\right)$$

$$\subset A^c \cup B^c \cup \left(\bigcup_i (C_i^c \cap E_i)\right) \qquad \text{(as } A \subset E_i\text{)}.$$

Thus

$$\mathbb{P}(u \text{ is not } D\text{-good}) \leqslant \mathbb{P}(A^c) + \mathbb{P}(B^c) + \sum_{i=1}^{n-1} \mathbb{P}(C_i^c \cap E_i).$$

Below, we bound separately the three terms of the right hand side.

(a) Denote $m' = \lfloor m/6 \rfloor$. For any $i \in \{0, \ldots, n-1\}$, the set $\mathcal{A}_i = \{j \in \{1, \ldots, m'\} \mid u_j \in [v_i, v_{i+1}]\}$ is empty with probability $(1 - (v_{i+1} - v_i))^{m'} \leqslant (1 - \Delta v)^{m'}$. Thus by the union bound, the probability that at least one of $\mathcal{A}_1, \ldots, \mathcal{A}_n$ is empty is upper bounded by $n(1 - \Delta v)^{m'}$.

We now check that $n(1 - \Delta v)^{m'} \leqslant \delta/18$. Indeed,

$$m' = \left\lfloor \frac{m}{6} \right\rfloor \geqslant \frac{m}{6} - 1 \geqslant \frac{3 + \log n + \log \frac{1}{\delta}}{\Delta v} \geqslant \frac{\log n + \log \frac{18}{\delta}}{\Delta v} \geqslant -\frac{\log n + \log \frac{18}{\delta}}{\log(1 - \Delta v)},$$

where we use $\Delta v \leqslant 1$, $3 \geqslant \log(18)$, and $\log(1 - \Delta v) \leqslant -\Delta v < 0$. This gives the desired inequality.

In other words, the probability that at least one of the intervals $[v_i, v_{i+1}]$ contains none of the $u_1, \ldots, u_{m'}$ is bounded by $\delta/18$. As a consequence, by the union bound, the probability that at least one of the intervals $[v_i, v_{i+1}]$ contains strictly less than 6 of the $u_1, \ldots, u_m$ is bounded by $\delta/3$, i.e., $\mathbb{P}(A^c) \leqslant \delta/3$.

(b) Recall that by convention, $u_0 = -\frac{\eta}{2}$ and $u_{m+1} = 1 + \frac{\eta}{2}$. For all $i \in \{0, \ldots, m+1\}$, denote $I_i = (u_i - D, u_i + D)$. Denote $F_j$ the event "$u_j \in I_i$ for some $i \in \{0, \ldots, m+1\}, i \neq j$". Note that $B^c = \cup_{j=1}^m F_j$.

Fix $j = 1, \ldots, m$. By conditioning on $u_i$ for all $i \in \{0, \ldots, m+1\}, i \neq j$, we see that $\mathbb{P}(F_j) \leqslant 2(m+1)D$. By the union bound,

$$\mathbb{P}(B^c) \leqslant 2m(m+1)D \leqslant \frac{\delta}{3}.$$

(c) Take $i \in \{1, \ldots, n-1\}$. For convenience, we define the random variable $u_i^{\mathrm{L}}$ (resp. $u_i^{\mathrm{R}}$) on the full probability space by setting $u_i^{\mathrm{L}} = 0$ (resp. $u_i^{\mathrm{R}} = 1$) when there is no neuron on the left (resp. the right) of $v_i$. We compute the joint cumulative distribution function of $(u_i^{\mathrm{L}}, u_i^{\mathrm{R}})$ (with a convenient change of inequality): for all $0 \leqslant y \leqslant v_i \leqslant z \leqslant 1$,

$$\mathbb{P}(u_i^{\mathrm{L}} \leqslant y, u_i^{\mathrm{R}} \geqslant z) = \mathbb{P}(\forall j \in \{1, \ldots, m\}, u_j \notin [y, z]) = (1 - (z - y))^m.$$

We observe that the joint cumulative distribution function of $(u_i^{\mathrm{L}}, u_i^{\mathrm{R}})$ is a smooth function of $(y, z)$ when $(y, z) \in (0, v_i) \times (v_i, 1)$. Note that the events $E_i$ and $\{(u_i^{\mathrm{L}}, u_i^{\mathrm{R}}) \in (0, v_i) \times (v_i, 1)\}$ are equal up to a null set. Therefore, on this event, $(u_i^{\mathrm{L}}, u_i^{\mathrm{R}})$ is an absolutely continuous random variable with density $g : (0, v_i) \times (v_i, 1) \to \mathbb{R}$,

$$g(y, z) = -\frac{\partial^2}{\partial y \partial z} \mathbb{P}(u_i^{\mathrm{L}} \leqslant y, u_i^{\mathrm{R}} \geqslant z) = m(m-1)(1 - (z - y))^{m-2}.$$

We compute

$$\mathbb{P}(C_i^c \cap E_i) = \mathbb{P}(\{|u_i^{\mathrm{R}} + u_i^{\mathrm{L}} - 2v_i| \leqslant D\} \cap E_i)$$

$$= \int_{\{0 < y < v_i < z < 1\}} m(m-1)(1 - (z - y))^{m-2} \mathbf{1}_{\{|y + z - 2v_i| \leqslant D\}} \mathrm{d}y \mathrm{d}z.$$

We make the change of variables $\theta = z - y$, $\nu = z + y$.

$$\mathbb{P}(C_i^c \cap E_i) = \frac{m(m-1)}{2} \int_{\{0 < \frac{\nu - \theta}{2} < v_i < \frac{\nu + \theta}{2} < 1\}} (1 - \theta)^{m-2} \mathbf{1}_{|\nu - 2v_i| \leqslant D} \mathrm{d}\theta \mathrm{d}\nu$$

$$\leqslant \frac{m(m-1)}{2} \left( \int_0^1 (1 - \theta)^{m-2} \mathrm{d}\theta \right) \left( \int_{-\infty}^\infty \mathbf{1}_{|\nu - 2v_i| \leqslant D} \mathrm{d}\nu \right)$$

$$= Dm.$$

Using $m \geqslant 24/\Delta v \geqslant 24n$, we have

$$\sum_{i=1}^{n-1} \mathbb{P}(C_i^c \cap E_i) \leqslant (n-1)Dm \leqslant \frac{\delta}{24 \times 6} \leqslant \frac{\delta}{3}.$$

This concludes the proof.

$\square$

# B Proofs of the results of the main text

## B.1 Proof of Proposition 1

Let us lower-bound the smallest eigenvalue of $H_\eta(u)$ which is equal to

$$\min_{\|a\|=1} a^\top H_\eta(u) a \,.$$

Now for $a \in \mathbb{R}^{m+1}$ such that $\|a\| = 1$,

$$a^\top H_\eta(u) a = \sum_{i,j=0}^m a_i a_j \int_0^1 \sigma_\eta(x - u_i) \sigma_\eta(x - u_j) \mathrm{d}x = \int_0^1 \left( \sum_{i=0}^m a_i \sigma_\eta(x - u_i) \right)^2 \mathrm{d}x \,.$$

Since $\Delta u > 2\eta$ (because $u \in \mathcal{U}$) and $u_0 = -\eta/2$, $u_{m+1} = 1 + \eta/2$, the intervals $[u_i + \eta/2, u_{i+1} - \eta/2]$ for $i \in \{0, \dots, m\}$ are disjoint and included in $[0, 1]$. Thus

$$a^\top H_\eta(u) a \geqslant \sum_{i=0}^m \int_{u_i + \eta/2}^{u_{i+1} - \eta/2} \left( \sum_{i=0}^m a_i \sigma_\eta(x - u_i) \right)^2 \mathrm{d}x \,.$$

Since $\sigma(x) = 0$ if $x < -1/2$ and $\sigma(x) = 1$ if $x > 1/2$, we have that $\sigma_\eta(x) = 0$ if $x < -\eta/2$ and $\sigma_\eta(x) = 1$ if $x > \eta/2$. Further recall that the $u_i$ are ordered in increasing order. As a consequence,

$$a^\top H_\eta(u) a \geqslant \sum_{i=0}^m \int_{u_i + \eta/2}^{u_{i+1} - \eta/2} \left( \sum_{k=0}^i a_k \right)^2 \mathrm{d}x$$

$$= \sum_{i=0}^m (u_{i+1} - u_i - \eta) \left( \sum_{k=0}^i a_k \right)^2$$

$$\geqslant \frac{\Delta(u)}{2} \sum_{i=0}^m \left( \sum_{k=0}^i a_k \right)^2 \,, \tag{12}$$

where in the last step, we used that $\Delta(u) > 2\eta$ and thus $u_{i+1} - u_i - \eta \geqslant \Delta(u) - \eta \geqslant \Delta(u) - \Delta(u)/2 = \Delta(u)/2$. Now, denote $c_0 = 0$ and $c_i = \sum_{k=0}^{i-1} a_k$. Then $\|a\| = 1$ writes

$$\sum_{i=0}^m (c_{i+1} - c_i)^2 = 1 \,.$$

Furthermore,

$$\sum_{i=0}^m (c_{i+1} - c_i)^2 = \sum_{i=0}^m c_{i+1}^2 + \sum_{i=0}^m c_i^2 - 2 \sum_{i=0}^m c_{i+1} c_i \leqslant 4 \sum_{i=0}^{m+1} c_i^2 \,.$$

Hence

$$\sum_{i=0}^{m+1} c_i^2 \geqslant \frac{1}{4} \,,$$

which shows in conjunction with (12) that the smallest eigenvalue of $H_\eta(u)$ is lower-bounded by $\frac{\Delta u}{8}$.

## B.2 Proof of Proposition 2

To show that $G(u) = (\nabla_u L_\eta)(a_\eta^*(u), u)$ is Lipschitz-continuous on $\mathcal{U}_\eta$, we show that it is differentiable on $\mathcal{U}_\eta$ and that its derivatives are uniformly bounded. The chain rule gives

$$\frac{\partial G_j}{\partial u_k} = \sum_{l=0}^m \frac{\partial a_{\eta,l}^*}{\partial u_k}(u) \frac{\partial^2 L_\eta}{\partial u_j \partial a_l}(a_\eta^*(u), u) + \frac{\partial^2 L_\eta}{\partial u_j \partial u_k}(a_\eta^*(u), u) \,.$$

From (10), using that $\sigma$ is twice continuously differentiable, it can be checked that $\frac{\partial L_\eta}{\partial u_j}$ is differentiable in both its arguments and its derivatives are uniformly upper-bounded when $a$ is bounded. Furthermore, for $u \in \mathcal{U}_\eta$,

$$\|a_\eta^*(u)\| \leqslant \|H_\eta(u)^{-1}\| \, \|b_\eta(u)\| \leqslant \frac{8M\sqrt{m+1}}{\Delta(u)},$$

by Lemma 1 and Proposition 1. Finally, according to Lemma 3, $a_\eta^*$ is differentiable with derivatives uniformly upper-bounded on $\mathcal{U}_\eta$. This concludes the proof.

### B.3 Proof of Proposition 3

In this proof, we denote $u_i^{\mathrm{L}}(\tau)$ (resp. $u_i^{\mathrm{R}}(\tau)$) the position at time $\tau$ of the neuron that is *at initialization* closest to $v_i$ to the left (resp. the right). Note that because of the movement of the neurons, it could be that $u_i^{\mathrm{L}}$ (resp. $u_i^{\mathrm{R}}$) does not remain the neuron closest to the left (resp. the right) throughout the dynamics. Our proof discusses when this phenomenon occurs. Similarly, denote $u_i^{\mathrm{LL}}$ (resp. $u_i^{\mathrm{RR}}$) the neuron second closest to the left (resp. the right) of $v_i$. Since the initialization is $D$-good, note that all these neurons are distinct.

Denote $\overline{\mathcal{T}}$ the minimal time $\tau \in [0, \mathcal{T}_{\max})$ such that $\Delta(u(\tau)) \leqslant D/2$ or there are less than two neurons in some piece $[v_i, v_{i+1}]$ of $f^*$. Note that by assumption, $\Delta(u(0)) \geqslant D > D/2$ and there are at least 6 neurons in each interval at initialization, thus $\overline{\mathcal{T}} > 0$. Furthermore, using the trivial inequalities $M \geqslant \Delta f/2$, $m + 1 \geqslant 1$ and $\eta^{1/2} \geqslant \eta$, we have $\frac{D}{2} = \frac{2^{11/2} M \sqrt{m+1} \sqrt{\eta}}{\Delta f} \geqslant 8\eta > 2\eta$. Recall that $2\eta$ is the quantity defining the set $\mathcal{U}_\eta$ supporting the maximal solution of the equation (8). As a consequence, we do have $\overline{\mathcal{T}} < \mathcal{T}_{\max}$. At the end of the proof, we check that $\mathcal{T} < \overline{\mathcal{T}}$, by controlling carefully the movement of each neuron.

Let us first bound the difference between the dynamics of $u$ and the dynamics that we would have if at each time $\tau$, the weights $a$ were given by $a_0^*(u(\tau))$, the best approximation of $f^*$ by a piecewise constant function with subdivision $u(\tau)$. For any $\tau < \overline{\mathcal{T}}$ and $j \in \{1, \dots, m\}$, by Lemma 6, we have

$$\left| \frac{\mathrm{d}u_j}{\mathrm{d}\tau}(\tau) + \frac{\partial L_\eta}{\partial u_j}(a_0^*(u(\tau)), u(\tau)) \right|$$
$$= \left| \frac{\partial L_\eta}{\partial u_j}(a_\eta^*(u(\tau)), u(\tau)) - \frac{\partial L_\eta}{\partial u_j}(a_0^*(u(\tau)), u(\tau)) \right|$$
$$\leqslant 2M(\sqrt{m+1} + 1)\|a_\eta^*(u(\tau)) - a_0^*(u(\tau))\| + \sqrt{m+1}\|a_\eta^*(u(\tau)) - a_0^*(u(\tau))\|^2. \quad (13)$$

We are therefore led to bounding $\|a_\eta^*(u(\tau)) - a_0^*(u(\tau))\|$, as follows:

$$\|a_\eta^*(u(\tau)) - a_0^*(u(\tau))\| \leqslant \frac{2^4 M \sqrt{m+1} \eta}{\Delta(u(\tau))} \qquad \text{(by Lemma 7)}$$
$$\leqslant \frac{2^5 M \sqrt{m+1} \eta}{D} \qquad \text{(since } \Delta(u(\tau)) \geqslant D/2 \text{)}$$
$$= \frac{D(\Delta f)^2}{2^8 M \sqrt{m+1}} \qquad \text{(by definition of } D \text{)}.$$

Then the first term in (13) is less than

$$\frac{(\sqrt{m+1} + 1)D(\Delta f)^2}{2^7 \sqrt{m+1}} \leqslant \frac{D(\Delta f)^2}{2^6},$$

and the second term in (13) is less than

$$\frac{D^2(\Delta f)^4}{2^{16} M^2 \sqrt{m+1}} \leqslant \frac{D(\Delta f)^2}{2^{14}}, \qquad \text{using } D \leqslant \Delta(u(0)) \leqslant 1, \Delta f \leqslant 2M \text{ and } m + 1 \geqslant 1.$$

Hence we obtain, for any $\tau < \overline{\mathcal{T}}$ and $j \in \{1, \dots, m\}$,

$$\left| \frac{\mathrm{d}u_j}{\mathrm{d}\tau}(\tau) + \frac{\partial L_\eta}{\partial u_j}(a_0^*(u(\tau)), u(\tau)) \right| \leqslant \frac{D(\Delta f)^2}{60} =: \Delta g \qquad (14)$$

Now, let us examine how the neurons move, by leveraging Lemma 4 that gives exact formulae for $\frac{\partial L_\eta}{\partial u_j}(a_0^*(u(\tau)), u(\tau))$. First, if $u_j$ is not next to a discontinuity, $\frac{\partial L_\eta}{\partial u_j}(a_0^*(u(\tau)), u(\tau)) = 0$, hence

$$|u_j(\tau) - u_j(0)| \leqslant (\Delta g)\tau.$$

Let us now study what happens next to a discontinuity $v_i$. Denote $(\delta f)_i = f_i^* - f_{i-1}^*$. W.l.o.g., assume that

$$u_i^{\mathrm{R}}(0) - v_i > v_i - u_i^{\mathrm{L}}(0).$$

In the reverse case, the proof is the same by swapping the roles of $u_i^{\mathrm{L}}$ and $u_i^{\mathrm{R}}$, and of $u_i^{\mathrm{LL}}$ and $u_i^{\mathrm{RR}}$. We are going to show that the dynamics of $u_i^{\mathrm{L}}$ are divided into two phases. Define $\mathcal{T}_i$ as the minimal $\tau \in [0, \overline{\mathcal{T}}]$ such that $u_i^{\mathrm{L}}(\tau) = v_i - \eta/2$. In the first phase $[0, \mathcal{T}_i]$, we have $u_i^{\mathrm{L}}(\tau) < v_i - \frac{\eta}{2}$ and $u_i^{\mathrm{L}}$ moves towards $v_i$. In the second phase $[\mathcal{T}_i, \overline{\mathcal{T}}]$, we show below that $u_i^{\mathrm{L}}(\tau) \in [v_i - \eta, v_i + \eta]$. Note that we can have $\mathcal{T}_i = 0$ if $u_i^{\mathrm{L}}(0) \geqslant v_i - \frac{\eta}{2}$. It is also possible that $\mathcal{T}_i = \infty$ a priori; this means that the second phase does not exist. We show below that this case does not happen. Figure 7 depicts the two phases.

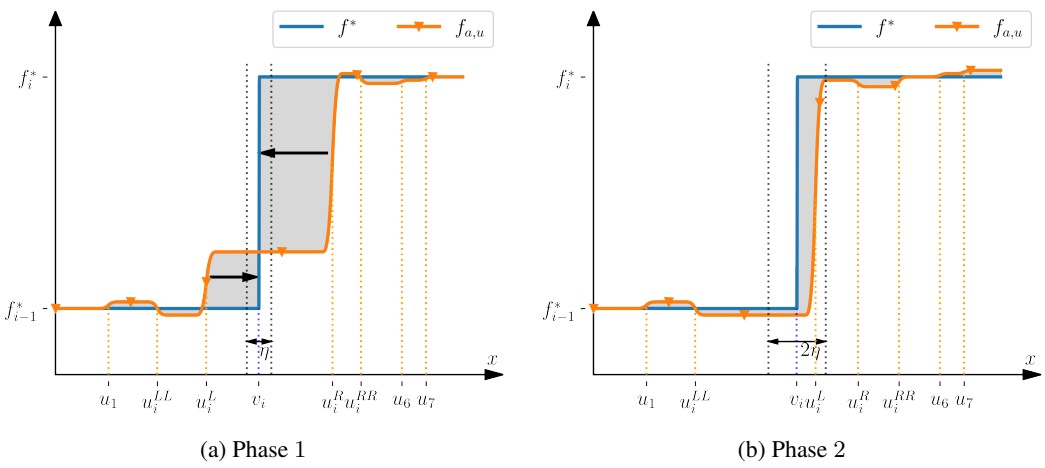

(a) Phase 1             (b) Phase 2

Figure 7: Dynamics of the neurons next to a discontinuity $v_i$. In the first phase, $u_i^{\mathrm{L}}$ and $u_i^{\mathrm{R}}$ move towards $v_i$, until the closest neuron (in this case $u_i^{\mathrm{L}}$) reaches the interval of size $\eta$ centered in $v_i$. In the second phase, $u_i^{\mathrm{L}}$ remains in an interval of size $2\eta$ around $v_i$, and none of the neurons move significantly.

Beginning by the first phase, we have, while $u_i^{\mathrm{L}}(\tau) < v_i - \frac{\eta}{2}$ and $u_i^{\mathrm{R}}(\tau) > v_i + \frac{\eta}{2}$, according to Lemma 4,

$$\frac{\partial L_\eta}{\partial u_i^{\mathrm{L}}}(a_0^*(u(\tau)), u(\tau)) = -\frac{1}{2}\frac{(u_i^{\mathrm{R}}(\tau) - v_i)^2(\delta f)_i^2}{(u_i^{\mathrm{R}}(\tau) - u_i^{\mathrm{L}}(\tau))^2},$$

$$\frac{\partial L_\eta}{\partial u_i^{\mathrm{R}}}(a_0^*(u(\tau)), u(\tau)) = \frac{1}{2}\frac{(v_i - u_i^{\mathrm{L}}(\tau))^2(\delta f)_i^2}{(u_i^{\mathrm{R}}(\tau) - u_i^{\mathrm{L}}(\tau))^2}.$$

For ease of computation, let $d_i^{\mathrm{L}}(\tau) = v_i - u_i^{\mathrm{L}}(\tau)$ and $d_i^{\mathrm{R}}(\tau) = u_i^{\mathrm{R}}(\tau) - v_i$ be the distances between the neurons and $v_i$. Then, by (14),

$$\frac{\mathrm{d}d_i^{\mathrm{R}}}{\mathrm{d}\tau}(\tau) + \frac{\mathrm{d}d_i^{\mathrm{L}}}{\mathrm{d}\tau}(\tau) \leqslant -\frac{1}{2}\frac{((d_i^{\mathrm{R}}(\tau))^2 + (d_i^{\mathrm{L}}(\tau))^2)(\delta f)_i^2}{(d_i^{\mathrm{L}}(\tau) + d_i^{\mathrm{R}}(\tau))^2} + 2\Delta g$$

$$\leqslant -\frac{(\Delta f)^2}{4} + 2\frac{D(\Delta f)^2}{60} \leqslant -\frac{(\Delta f)^2}{5}$$

since $D \leqslant \Delta(u(0)) \leqslant 1$. Thus, in some time less than $\mathcal{T} = \frac{6}{(\Delta f)^2}$, $d_i^{\mathrm{R}}(\tau) + d_i^{\mathrm{L}}(\tau) \leqslant \eta$, that is, either $u_i^{\mathrm{L}}$ reaches $v_i - \frac{\eta}{2}$ or $u_i^{\mathrm{R}}$ reaches $v_i + \frac{\eta}{2}$. Let us check that the second event cannot actually

happen: while $u_i^L(\tau) < v_i - \frac{\eta}{2}$ and $u_i^R(\tau) > v_i + \frac{\eta}{2}$, we also have

$$\frac{\mathrm{d}d_i^R}{\mathrm{d}\tau}(\tau) - \frac{\mathrm{d}d_i^L}{\mathrm{d}\tau}(\tau) \geqslant \frac{((d_i^R(\tau))^2 - (d_i^L(\tau))^2)(\delta f)_i^2}{(d_i^L(\tau) + d_i^R(\tau))^2} - 2\Delta g$$
$$= \frac{(d_i^R(\tau) - d_i^L(\tau))(\delta f)_i^2}{d_i^L(\tau) + d_i^R(\tau)} - 2\Delta g\,.$$

By condition (c) of Definition 2 and by (14), we have $d_i^R(0) - d_i^L(0) \geqslant D = \frac{60\Delta g}{(\Delta f)^2} \geqslant \frac{60\Delta g}{(\delta f)_i^2}$, and furthermore $d_i^L(\tau) + d_i^R(\tau) \leqslant 1$. An easy reasoning then shows that $d_i^R - d_i^L$ is increasing. Therefore $u_i^R$ must remain further away from $v_i$ than $u_i^L$.

In summary, we showed that there exists some time $\mathcal{T}_i \leqslant \mathcal{T}$ when $u_i^L(\mathcal{T}_i) = v_i - \frac{\eta}{2}$, which marks the end of the first phase, and we also have

$$d_i^R(\mathcal{T}_i) - d_i^L(\mathcal{T}_i) \geqslant d_i^R(0) - d_i^L(0) \geqslant D\,.$$

Moving on to the study of the second phase, let us show that $u_i^L(\tau)$ stays in the interval $[v_i - \eta, v_i + \eta]$ for $\tau \in [\mathcal{T}_i, \overline{\mathcal{T}})$. Consider any $\tau \leqslant \overline{\mathcal{T}}$ such that $u_i^L(\tau) = v_i - \eta$. Then we have by (14) and Lemma 4

$$\frac{\mathrm{d}u_i^L}{\mathrm{d}\tau}(\tau) \geqslant \frac{(u_i^R(\tau) - v_i)^2(\delta f)_i^2}{(u_i^R(\tau) - v_i + \eta)^2} - \Delta g \geqslant \Delta g\,, \tag{15}$$

where the second upper bound comes from the fact that we have $u_i^R(\tau) - v_i \geqslant \frac{D}{2} - \eta$ since $\Delta(u(\tau)) \geqslant D/2$, and furthermore, $x \mapsto \frac{x^2}{(x+\eta)^2}$ is increasing, hence

$$\frac{(u_i^R - v_i)^2(\delta f)_i^2}{(u_i^R - v_i + \eta)^2} \geqslant \left(\frac{\frac{D}{2} - \eta}{\frac{D}{2}}\right)^2 \Delta f^2 \underset{(D/2 \geqslant 2\eta)}{\geqslant} \frac{(\Delta f)^2}{4} \geqslant 2\Delta g\,.$$

Equation (15) implies that $u_i^L(\tau) \geqslant v_i - \eta$ for all $\tau \in [\mathcal{T}_i, \overline{\mathcal{T}})$. Similarly, consider any $\tau \leqslant \overline{\mathcal{T}}$ such that $u_i^L(\tau) = v_i + \eta$. Note that, for such a $\tau$, $u_i^L(\tau)$ is now on the right of $v_i$, and the neurons flanking $v_i$ are $u_i^{LL}$ and $u_i^L$. Thus we have by (14) and Lemma 4

$$\frac{\mathrm{d}u_i^L}{\mathrm{d}\tau}(\tau) \leqslant -\frac{(v_i - u_i^{LL}(\tau))^2(\delta f)_i^2}{(v_i + \eta - u_i^{LL}(\tau))^2} + \Delta g \leqslant -\Delta g\,,$$

where the second lower bound unfolds similarly as previously. This shows that $u_i^L(\tau) \leqslant v_i + \eta$ for all $\tau \in [\mathcal{T}_i, \overline{\mathcal{T}})$.

We now check that $\mathcal{T} < \overline{\mathcal{T}}$, that is, for all $\tau \leqslant \mathcal{T}$, $\Delta(u(\tau)) > D/2$ and there are at least two neurons in each interval $[v_i, v_{i+1}]$. Starting with the first condition, we say that neurons $u_j$ and $u_k$ collide if $|u_j(\tau) - u_k(\tau)| = D/2$ for some $\tau \leqslant \mathcal{T}$. Let us show that no pair of neurons collide.

We start by showing that there is no collision between $u_i^{LL}$ and $u_i^L$. In the first phase $[0, \mathcal{T}_i]$, we have $\frac{\mathrm{d}u_i^{LL}}{\mathrm{d}\tau}(\tau) \leqslant \Delta g$. Recall that we also have $\frac{\mathrm{d}u_i^L}{\mathrm{d}\tau}(\tau) \geqslant -\Delta g$ and thus for $\tau \leqslant \mathcal{T}_i$,

$$u_i^L(\tau) - u_i^{LL}(\tau) \geqslant u_i^L(0) - u_i^{LL}(0) - 2\mathcal{T}\Delta g \geqslant \frac{4D}{5}$$

since $u_i^L(0) - u_i^{LL}(0) \geqslant D$ and $\mathcal{T}\Delta g = D/10$ by definition of $\mathcal{T}$ and $\Delta g$. As a consequence, $u_i^{LL}$ and $u_i^L$ do not collide during the first phase, and we have

$$u_i^{LL}(\mathcal{T}_i) \leqslant u_i^L(\mathcal{T}_i) - \frac{4D}{5} = v_i - \frac{\eta}{2} - \frac{4D}{5}\,. \tag{16}$$

In the second phase, we can have $u_i^L \in [v_i, v_i + \eta]$ in which case $u_i^{LL}$ becomes the neuron flanking $v_i$ to the left and $u_i^L$ the neuron flanking to the right. Then (14) and Lemma 4 give

$$\frac{\mathrm{d}u_i^{LL}}{\mathrm{d}\tau} \leqslant \frac{(u_i^L(\tau) - v_i)^2(\delta f)_i^2}{(u_i^L(\tau) - u_i^{LL}(\tau))^2} + \Delta g \leqslant \frac{16\eta^2 M^2}{D^2} + \Delta g\,.$$

Of course, this bound also holds when $u_i^{\mathrm{L}} \in [v_i - \eta, v_i]$, because then $\frac{\mathrm{d} u_i^{\mathrm{LL}}}{\mathrm{d}\tau} \leqslant \Delta g$. Thus, in the second phase $\tau \in [\mathcal{T}_i, \mathcal{T}]$, by the previous upperbound and the fact that $u_i^{\mathrm{L}}(\tau) \geqslant v_i - \frac{\eta}{2}$,

$$u_i^{\mathrm{L}}(\tau) - u_i^{\mathrm{LL}}(\tau) \geqslant v_i - \frac{\eta}{2} - \left( u_i^{\mathrm{LL}}(\mathcal{T}_i) + (\tau - \mathcal{T}_i)\left( \frac{16\eta^2 M^2}{D^2} + \Delta g \right) \right)$$
$$\geqslant \frac{4D}{5} - \mathcal{T}\left( \frac{16\eta^2 M^2}{D^2} + \Delta g \right),$$

by (16). Let us now upper-bound each of the last two terms by $D/10$ to conclude. By definition of $D$,

$$\eta = \frac{(\Delta f)^2 D^2}{2^{13}(m+1)M^2}.$$

Thus

$$\frac{16\eta^2 M^2 \mathcal{T}}{D^2} = \frac{3(\Delta f)^2 D^2}{2^{21}(m+1)^2 M^2} \leqslant \frac{D}{10}$$

using the definition of $\mathcal{T}$, $D \leqslant \Delta(u(0)) \leqslant 1$, $\Delta f \leqslant 2M$ and $m+1 \geqslant 1$. Finally, $\mathcal{T}\Delta g = D/10$. Thus $u_i^{\mathrm{LL}}$ and $u_i^{\mathrm{L}}$ do not collide.

We now show that $u_i^{\mathrm{L}}$ and $u_i^{\mathrm{R}}$ do not collide. In the first phase $\tau \in [0, \mathcal{T}_i]$, we have

$$u_i^{\mathrm{R}}(\tau) - u_i^{\mathrm{L}}(\tau) \geqslant u_i^{\mathrm{R}}(\tau) - v_i = d_i^{\mathrm{R}}(\tau) \geqslant d_i^{\mathrm{R}}(\tau) - d_i^{\mathrm{L}}(\tau) \geqslant D.$$

As a consequence, $u_i^{\mathrm{L}}$ and $u_i^{\mathrm{R}}$ do not collide during the first phase, and we have

$$u_i^{\mathrm{R}}(\mathcal{T}_i) \geqslant D + u_i^{\mathrm{L}}(\mathcal{T}_i) = D + v_i - \frac{\eta}{2}. \tag{17}$$

In the second phase, $u_i^{\mathrm{R}}$ plays a role symmetric to $u_i^{\mathrm{LL}}$: it can be, or not, the neuron closest to the right of $v_i$, depending on whether $u_i^{\mathrm{L}} \in [v_i - \eta, v_i]$ or $u_i^{\mathrm{L}} \in [v_i, v_i + \eta]$. As for $u_i^{\mathrm{LL}}$, we can show that in any case, for $\tau \in [\mathcal{T}_i, \mathcal{T}]$,

$$\frac{\mathrm{d} u_i^{\mathrm{R}}}{\mathrm{d}\tau} \geqslant -\frac{16\eta^2 M^2}{D^2} - \Delta g.$$

Thus one concludes as before: for $\tau \in [\mathcal{T}_i, \mathcal{T}]$, by the previous lowerbound and the fact that $u_i^{\mathrm{L}}(\tau) \leqslant v_i + \frac{\eta}{2}$,

$$u_i^{\mathrm{R}}(\tau) - u_i^{\mathrm{L}}(\tau) \geqslant u_i^{\mathrm{R}}(\mathcal{T}_i) - (\tau - \mathcal{T}_i)\left( \frac{16\eta^2 M^2}{D^2} + \Delta g \right) - \left( v_i + \frac{\eta}{2} \right).$$

Then, by (17),

$$u_i^{\mathrm{R}}(\tau) - u_i^{\mathrm{L}}(\tau) \geqslant D - \eta - \mathcal{T}\left( \frac{16\eta^2 M^2}{D^2} + \Delta g \right) > \frac{D}{2},$$

where the last lower-bound unfolds similarly as for $u_i^{\mathrm{LL}}$ and $u_i^{\mathrm{L}}$. Thus there is no collision between $u_i^{\mathrm{L}}$ and $u_i^{\mathrm{R}}$.

The reader can check that all other pairs of neurons do not collide, including those involving $u_0 = -\eta/2$ and $u_{m+1} = 1 + \eta/2$. In fact, the proof is easier than for $u_i^{\mathrm{LL}}, u_i^{\mathrm{L}}$ and $u_i^{\mathrm{L}}, u_i^{\mathrm{R}}$ because the discontinuity at $v_i$ attracts these neurons together.

Furthermore, we proved that before time $\mathcal{T}$ at most one neuron can escape on each side of a piece $[v_i, v_{i+1}]$ of $f$. Since we start with at least four (and even six) neurons per piece, there is always before $\mathcal{T}$ at least two neurons per piece.

This shows that $\mathcal{T} < \overline{\mathcal{T}}$, and we also proved that at time $\mathcal{T}$, all discontinuities have finished their first phase, hence there is a neuron at distance less than $\eta$ from each discontinuity of the target function.

### B.4 Proof of Theorem 2

Take $C = 2^{-19}$. Then by assumption of Theorem 2,

$$\eta \leqslant \frac{\delta^2 (\Delta f)^2}{2^{19} M^2 (m+1)^5}.$$

Moreover, by the definition of $D$ from Proposition 3,

$$\eta = \frac{(\Delta f)^2 D^2}{2^{13} M^2 (m+1)} \,.$$

This implies that

$$D^2 \leqslant \frac{\delta^2}{2^6 (m+1)^4} \,,$$

and in consequence

$$D \leqslant \frac{\delta}{6(m+1)^2} \,.$$

Then Lemma 10 shows that the initialization is $D$-good with probability at least $1 - \delta$ (since the $D$-good property is monotonous in $D$).

Hence, with probability at least $1 - \delta$, according to Proposition 3, the maximal solution to (8) is defined at least until $\mathcal{T}$ and at that time, there is a neuron at distance less than $\eta$ from each discontinuity of the target function. Furthermore, $3\eta \leqslant \frac{1}{m+1} \leqslant \frac{1}{n} \leqslant \Delta v$, hence Lemma 9 applies. This implies that

$$\int_0^1 |f^*(x) - f(x; a^*(u(\mathcal{T})), u(\mathcal{T}))|^2 \mathrm{d}x \leqslant 6M^2 \eta n \,.$$

The assumption on $\eta$ allow to conclude that the upper-bound is less than $\xi$.

**Remark 2.** *We did not try to optimize the value of $C$ since our goal was to show convergence to a global optimum and the dependency of the dynamics on the parameters (for instance, it is remarkable that $\mathcal{T}$ does not depend on $\xi$).*

### B.5 Proof of Proposition 4

For $s \leqslant t$, Proposition 1 holds since for $\Delta(u(s)) \geqslant 16\eta > 2\eta$. Thus $a_\eta^*(u(s))$ is well-defined and verifies

$$\nabla_a L_\eta(a_\eta^*(u(s)), u(s)) = 0 \,.$$

Let, for $s \leqslant t$, $V(s) = \|a(s) - a_\eta^*(u(s))\|$. Recall that, by (11),

$$\nabla_a L_\eta(a, u) = H_\eta(u)a - b_\eta(u) \,.$$

Hence, for $s \leqslant t$,

$$
\begin{aligned}
\langle a(s) - a_\eta^*(u(s)), &\nabla_a L_\eta(a(s), u(s)) \rangle \\
&= \langle a(s) - a_\eta^*(u(s)), \nabla_a L_\eta(a(s), u(s)) - \nabla_a L_\eta(a_\eta^*(u(s)), u(s)) \rangle \\
&= \langle a(s) - a_\eta^*(u(s)), H_\eta(u(s))(a(s) - a_\eta^*(u(s))) \rangle \\
&\geqslant \frac{\Delta(u(s))}{8} V(s)^2 \\
&\geqslant \frac{D}{16} V(s)^2 \,,
\end{aligned}
$$

where the first lower bound is a consequence of Proposition 1. Then we have, for any $s \leqslant t$,

$$
\begin{aligned}
\frac{\mathrm{d}}{\mathrm{d}s}\left(\frac{1}{2} V(s)^2\right) &= \left\langle a(s) - a_\eta^*(u(s)), \frac{\mathrm{d}a}{\mathrm{d}s}(s) - \frac{\mathrm{d}}{\mathrm{d}s} a_\eta^*(u(s)) \right\rangle \\
&= \left\langle a(s) - a_\eta^*(u(s)), -\nabla_a L_\eta(a(s), u(s)) - \frac{\mathrm{d}}{\mathrm{d}s} a_\eta^*(u(s)) \right\rangle \\
&\leqslant -\frac{D}{16} V(s)^2 + \left\| \frac{\mathrm{d}}{\mathrm{d}s} a_\eta^*(u(s)) \right\| V(s) \,.
\end{aligned}
$$

Let us now upper bound the norm appearing in the second term. We first have by the chain rule

$$\frac{\mathrm{d}}{\mathrm{d}s} a_\eta^*(u(s)) = \frac{\partial a_\eta^*}{\partial u}(u(s)) \frac{\mathrm{d}u}{\mathrm{d}s}(s) \,.$$

By Lemma 3 (which holds since for $\Delta(u(s)) \geqslant 16\eta > 2\eta$),

$$\left\|\frac{\partial a_\eta^*}{\partial u}(u(s))\right\| \leqslant \frac{8}{\Delta(u(s))}\left(2(m+1)\|a_\eta^*(u(s))\| + M\right).$$

Besides,

$$\left\|\frac{\mathrm{d}u}{\mathrm{d}s}(s)\right\| \leqslant \varepsilon\|\nabla_u L_\eta(a(s), u(s))\|.$$

By Lemma 8,

$$\|\nabla_u L_\eta(a(s), u(s))\| \leqslant \sqrt{m+1}\|a(s)\|^2 + M\|a(s)\|. \tag{18}$$

Furthermore,

$$\|a(s)\| \leqslant \|a_\eta^*(u(s))\| + \|a(s) - a_\eta^*(u(s))\| = \|a_\eta^*(u(s))\| + V(s).$$

By Lemmas 6 and 7, which apply since $\Delta(u(s)) > 2\eta$ and since there are at least two positions $u_j(s)$ in each interval $[v_i, v_{i+1}]$ for $s \leqslant t$,

$$\begin{aligned}
\|a_\eta^*(u(s))\| &\leqslant \|a_0^*(u(s))\| + \|a_0^*(u(s)) - a_\eta^*(u(s))\| \\
&\leqslant 2M\sqrt{m+1} + \frac{16M\sqrt{m+1}\eta}{\Delta(u(s))} \\
&\leqslant 2M\sqrt{m+1} + \frac{32M\sqrt{m+1}\eta}{D} \\
&\leqslant 3M\sqrt{m+1},
\end{aligned}$$

where the last upper bound is implied by the assumption $D \geqslant 32\eta$.

Now define $T_{\max} = \inf\left\{s \geqslant 0, V(s) > 3M\sqrt{m+1}\right\}$ and assume $s \leqslant \min(t, T_{\max})$ so that $V(s) \leqslant 3M\sqrt{m+1}$. Then we proved that $\|a(s)\| \leqslant 6M\sqrt{m+1}$. Going back to (18), we deduce that

$$\|\nabla_u L_\eta(a(s), u(s))\| \leqslant 36M^2(m+1)^{3/2} + 6M^2\sqrt{m+1} \leqslant 2^6 M^2(m+1)^{3/2}. \tag{19}$$

Putting everything together, we obtain

$$\left\|\frac{\mathrm{d}}{\mathrm{d}s}a_\eta^*(u(s))\right\| \leqslant \frac{2^9 M^2(m+1)^{3/2}}{\Delta(u(s))}\left(6M(m+1)^{3/2} + M\right)\varepsilon \leqslant \frac{2^{13}M^3(m+1)^3}{D}\varepsilon.$$

All in all,

$$\frac{\mathrm{d}}{\mathrm{d}s}\left(\frac{1}{2}V(s)^2\right) \leqslant -\frac{D}{16}V(s)^2 + \frac{2^{13}M^3(m+1)^3}{D}\varepsilon V(s).$$

Hence

$$\frac{\mathrm{d}}{\mathrm{d}s}(V(s)) = \frac{1}{V(s)}\frac{\mathrm{d}}{\mathrm{d}s}\left(\frac{1}{2}V(s)^2\right) \leqslant -\frac{D}{16}V(s) + \frac{2^{13}M^3(m+1)^3}{D}\varepsilon.$$

By Grönwall's inequality, for all $s \leqslant \min(t, T_{\max})$,

$$V(s) \leqslant \exp^{-\frac{D}{16}s} V(0) + \frac{2^{17}M^3(m+1)^3}{D^2}\varepsilon(1 - \exp^{-\frac{D}{16}s}) \tag{20}$$

$$\leqslant \exp^{-\frac{D}{16}s} V(0) + \frac{2^{17}M^3(m+1)^3}{D^2}\varepsilon. \tag{21}$$

Finally note that $V(0) = \|a_\eta^*(0)\| \leqslant 2M\sqrt{m+1}$ and $\frac{2^{17}M^3(m+1)^3\varepsilon}{D^2} \leqslant 2M\sqrt{m+1}$ by the assumption of the Proposition on $\varepsilon$. Hence (20) implies that for all $s \leqslant \min(t, T_{\max})$, $V(s)$ is a (weighted) average of two terms less than $2M\sqrt{m+1}$ hence it is less than $2M\sqrt{m+1}$. This shows that $T_{\max} \geqslant t$, which concludes the proof since (21) is then valid for $s = t$.

## B.6 Proof of Theorem 1

In the proof, we take $C_1 = 2^{-21}$ and $C_2 = 2^{-36}$. Denote

$$D = \frac{\delta}{6(m+1)^2}.$$

Lemma 10 shows that the initialization is $D$-good with probability at least $1 - \delta$. In the following, we study the case where this event happens.

Denote $\overline{T}$ the minimal time $t > 0$ such that $\Delta(u(t)) \leqslant D/2$ or there are less than two neurons in some piece $[v_i, v_{i+1}]$ of $f^*$ or $\|a(t)\| > 7M\sqrt{m+1}$. Note that $\overline{T} > 0$ since the initialization is $D$-good. By Lemma 8, $\nabla_u L_\eta$ and $\nabla_a L_\eta$ are Lipschitz-continuous on compacts, hence the solution of the gradient flow is well defined for $t < \overline{T}$ since $\overline{T}$ defines a compact set of parameters.

Then all the assumptions of Proposition 4 are satisfied on the time interval $[0, t]$ for any $t < \overline{T}$. More precisely, the assumptions that do not come directly from the definition of $\overline{T}$ are the lower bound for $D$ and the upper bound for $\varepsilon$. The lower bound for $D$ come from

$$D = \frac{\delta}{6(m+1)^2} \geqslant 32\eta \tag{22}$$

by (3) and the simple bounds $\delta \leqslant 1$, $\Delta f \leqslant 2M$, $m + 1 \geqslant 1$. The upper bound for $\varepsilon$ comes from (3) since

$$\varepsilon \leqslant \frac{\delta^3 (\Delta f)^2}{2^{36} M^4 (m+1)^{19/2}} \leqslant \frac{\delta^2}{36 \cdot 2^{16} M^2 (m+1)^{13/2}} = \frac{D^2}{2^{16} M^2 (m+1)^{5/2}},$$

where the second upper bound uses $m \geqslant 0$, $\delta \leqslant 1$ and $\Delta f \leqslant 2M$. Therefore, according to Proposition 4,

$$\|a(t) - a_\eta^*(u(t))\| \leqslant 3M\sqrt{m+1} \exp^{-\frac{D}{16} t} + \frac{2^{17} M^3 (m+1)^3}{D^2} \varepsilon, \tag{23}$$

Furthermore, the proof of Proposition 4 actually implies that

$$\|a_\eta^*(u(t))\| \leqslant 3M\sqrt{m+1} \quad \text{and} \quad \|a(t)\| \leqslant 6M\sqrt{m+1}. \tag{24}$$

The second bound implies that the condition $\|a(t)\| > 7M\sqrt{m+1}$ in the definition of $\overline{T}$ is actually never active. Let us distinguish between two phases: letting

$$T_0 = \frac{16}{D} \log\left(\frac{2^{16} M^2 (m+1)^3}{\delta (\Delta f)^2}\right) = \frac{96(m+1)^2}{\delta} \log\left(\frac{2^{16} M^2 (m+1)^3}{\delta (\Delta f)^2}\right),$$

then the first phase corresponds to $t \leqslant T_0$ and the second phase for $t \geqslant T_0$.

**Analysis of the first phase.** In the first phase, each neuron moves at most by

$$\varepsilon T_0 \max_j \left| \frac{\partial L_\eta}{\partial u_j}(a(t), u(t)) \right| \leqslant \varepsilon T_0 \|\nabla_u L_\eta(a(s), u(s))\| \leqslant 2^6 \varepsilon T_0 M^2 (m+1)^{3/2},$$

where the second upper bound comes from (19) in the proof of Proposition 4. This quantity is less than $\frac{D}{8}$ if

$$\frac{6144(m+1)^{7/2} M^2}{\delta} \log\left(\frac{2^{16} M^2 (m+1)^3}{\delta (\Delta f)^2}\right) \varepsilon \leqslant \frac{\delta}{48(m+1)^2}.$$

Let us check this condition: we have

$$\frac{6144(m+1)^{7/2} M^2}{\delta} \log\left(\frac{2^{16} M^2 (m+1)^3}{\delta (\Delta f)^2}\right) \varepsilon$$

$$= \frac{16 \cdot 6144(m+1)^{7/2} M^2}{\delta} \log\left(\frac{2 M^{1/8} (m+1)^{3/16}}{\delta^{1/16} (\Delta f)^{1/8}}\right) \varepsilon$$

$$\leqslant \frac{16 \cdot 6144(m+1)^{7/2} M^2}{\delta} \log\left(\frac{4M(m+1)}{\delta \Delta f}\right) \varepsilon,$$

since $m + 1 \geqslant 1$, $\delta \leqslant 1$, and $2M/\Delta f \geqslant 1$, hence $(2M/\Delta f)^{1/8} \leqslant 2M/\Delta f$. Next, upper-bounding $\log(x)$ by $x$, we have, by (3),

$$\frac{768(m+1)^{7/2}M^2}{\delta} \log\left(\frac{2^{16}M^2(m+1)^3}{\delta(\Delta f)^2}\right)\varepsilon \leqslant \frac{64 \cdot 6144(m+1)^{9/2}M^3}{\delta^2 \Delta f}\varepsilon$$

$$\leqslant \frac{6144\delta(\Delta f)}{2^{29}M(m+1)^5}$$

$$\leqslant \frac{\delta}{48(m+1)^2}$$

using $\Delta f \leqslant 2M$ and $m \geqslant 0$. Note that the upper bound $2^6 \varepsilon T_0 M^2(m+1)^{3/2} \leqslant D/8$ also implies that

$$T_0 \leqslant \frac{D}{2^9 \varepsilon M^2(m+1)^{3/2}} \leqslant \frac{1}{2\varepsilon(\Delta f)^2} = \frac{T}{12} \tag{25}$$

since $m \geqslant 0$, $D \leqslant 1$ and $\Delta f \leqslant 2M$. Since each neuron moves by at most $D/8$ in the time interval $[0, T_0]$ and since $\Delta(u(0)) \geqslant D$, we deduce that

$$\Delta(u(T_0)) \geqslant \frac{3}{4}D. \tag{26}$$

Similarly, by condition (c) of the definition of a $D$-good vector, for all discontinuities $v_i$,

$$|u_i^{\mathrm{R}}(0) + u_i^{\mathrm{L}}(0) - 2v_i| \geqslant D,$$

thus

$$|u_i^{\mathrm{R}}(T_0) + u_i^{\mathrm{L}}(T_0) - 2v_i| \geqslant \frac{3}{4}D. \tag{27}$$

Furthermore, there at least four neurons on each piece of $f$ at $T_0$, because at most one neuron can move out of each piece by either side between $0$ and $T_0$.

**Analysis of the second phase.** Let

$$\Delta a = \frac{D(\Delta f)^2}{2^9 M\sqrt{m+1}} = \frac{\delta(\Delta f)^2}{6 \cdot 2^9 M(m+1)^{5/2}}.$$

In the second phase $t \geqslant T_0$, we are able to control by $\Delta a$ the distance between $a(t)$ and the weights $a_0^*(u(t))$ that are the best approximation of $f^*$ by a piecewise affine function with subdivision $u(t)$. To show this, first note that the first term in (23) is smaller than $\frac{\Delta a}{4}$ when

$$3M\sqrt{m+1}\exp^{-\frac{D}{16}t} \leqslant \frac{\Delta a}{4}.$$

which is equivalent to

$$t \geqslant \log\left(\frac{12M\sqrt{m+1}}{\Delta a}\right)\frac{16}{D}.$$

which is implied by $t \geqslant T_0$. Furthermore, the second term in (23) is smaller than $\frac{\Delta a}{4}$ because, by definition of $D$ and by (3),

$$\frac{2^{17}M^3(m+1)^3}{D^2}\varepsilon = \frac{36 \cdot 2^{17}M^3(m+1)^7}{\delta^2}\varepsilon \leqslant \frac{6^2\delta(\Delta f)^2}{2^{19}M(m+1)^{5/2}} = \frac{6^3\Delta a}{2^{10}} \leqslant \frac{\Delta a}{4}.$$

Hence, for all $T_0 \leqslant t < \overline{T}$,

$$\|a(t) - a_\eta^*(u(t))\| \leqslant \frac{\Delta a}{2}.$$

Furthermore, note that the assumption of Lemma 7 applies for $t < \overline{T}$ since $\Delta(u(t)) \geqslant \frac{D}{2} > 2\eta$ by (22). Therefore, by Lemma 7 and by (3),

$$
\begin{aligned}
\|a_\eta^*(u(t)) - a_0^*(u(t))\| &\leqslant \frac{2^4 M \sqrt{m+1}}{\Delta(u(t))} \eta \\
&\leqslant \frac{2^5 M \sqrt{m+1}}{D} \eta \\
&= \frac{2^5 \cdot 6 M (m+1)^{5/2}}{\delta} \eta \\
&\leqslant \frac{6\delta(\Delta f)^2}{2^{16} M (m+1)^{5/2}} \\
&= \frac{6^2 \Delta a}{2^7} \leqslant \frac{\Delta a}{2} \, .
\end{aligned}
$$

By the triangular inequality, we deduce the upper bound that we were after, that is

$$
\|a(t) - a_0^*(u(t))\| \leqslant \Delta a \, .
$$

As in the proof of Proposition 3, we can now control the distance between the true dynamics and the one that we would have if the weights were equal to $a_0^*(u)$. Namely, for any $T_0 \leqslant t \leqslant \overline{T}$ and $j \in \{1, \ldots, m\}$, by Lemma 6 (which applies since $\Delta(u(t)) > 2\eta$ by (22)), we have

$$
\begin{aligned}
&\left| \frac{\mathrm{d}u_j}{\mathrm{d}t}(t) + \frac{\partial L_\eta}{\partial u_j}(a_0^*(u(t)), u(t)) \right| \\
&= \left| \frac{\partial L_\eta}{\partial u_j}(a(t), u(t)) - \frac{\partial L_\eta}{\partial u_j}(a_0^*(u(t)), u(t)) \right| \\
&\leqslant 2M(\sqrt{m+1}+1)\|a(t) - a_0^*(u(t))\| + \sqrt{m+1}\|a(t) - a_0^*(u(t))\|^2 \, .
\end{aligned}
$$

The first term is less than

$$
2M(\sqrt{m+1}+1)\Delta a = \frac{(\sqrt{m+1}+1)D(\Delta f)^2}{2^8 \sqrt{m+1}} \leqslant \frac{D(\Delta f)^2}{2^7} \, ,
$$

and the second term is less than

$$
\sqrt{m+1}(\Delta a)^2 = \frac{D^2(\Delta f)^4}{2^{18} M^2 \sqrt{m+1}} \leqslant \frac{D(\Delta f)^2}{2^{16}} \, ,
$$

using $D \leqslant \Delta(u(0)) \leqslant 1$, $\Delta f \leqslant 2M$ and $m+1 \geqslant 1$. Hence we obtain, for any $T_0 \leqslant t \leqslant \overline{T}$ and $j \in \{1, \ldots, m\}$,

$$
\left| \frac{\mathrm{d}u_j}{\mathrm{d}t}(t) + \frac{\partial L_\eta}{\partial u_j}(a_0^*(u(t)), u(t)) \right| \leqslant \frac{D(\Delta f)^2}{120} \, .
$$

We are therefore in a situation very similar to the proof of Proposition 3, starting from (14). One can check that all the arguments used in the proof also apply here. On top of the estimate above that resembles (14), the crucial facts that make the argument of Proposition 3 work here are the bounds (26) and (27) as well as the fact that there are at least four neurons on each piece $f$ at $T_0$, which together are very similar to the conditions ensuring that $u(0)$ is $D$-good in the proof of Proposition 3. Another key point is (25), ensuring that a time at least equal to $11T/12$ remains after the first phase of this proof, which is enough time for the dynamics described in the proof of Proposition 3 to unfold.

This yields that $T < \overline{T}$, and that at time $T$, there is a neuron at distance less than $\eta$ from each discontinuity of $f^*$. Furthermore, $3\eta \leqslant \frac{1}{m+1} \leqslant \frac{1}{n} \leqslant \Delta v$, hence Lemma 9 applies. Thus

$$
\int_0^1 (f_\eta(x; a_\eta^*(u(T)), u(T)) - f^*(x))^2 \mathrm{d}x \leqslant 6M^2 \eta n \leqslant \frac{\xi}{2} \, ,
$$

where the second upper bound comes from $n \leqslant m+1$ and from (3). Furthermore, by (24) and by Lemma 8,

$$
\begin{aligned}
|L_\eta(a(T), u(T)) - L_\eta(a_\eta^*(u(T)), u(T))| &\leqslant \sqrt{m+1}(6M(m+1) + M)\|a(T) - a_\eta^*(u(T))\| \\
&\leqslant 16M(m+1)^{3/2}\|a(T) - a_\eta^*(u(T))\| \, .
\end{aligned}
$$

Let us show that this term is less than $\xi/4$. Recall that, by (23),

$$\|a(T) - a_\eta^*(u(T))\| \leqslant 3M\sqrt{m+1}\exp^{-\frac{D}{16}T} + \frac{2^{17}M^3(m+1)^3}{D^2}\varepsilon\,.$$

By definition of $D$ and $T$, by using $\exp(-x) \leqslant 1/x$ for $x \geqslant 1$ and by (3),

$$
\begin{aligned}
16M(m+1)^{3/2} \cdot 3M\sqrt{m+1}\exp^{-\frac{D}{16}T} &= 48M^2(m+1)^2\exp\left(-\frac{\delta}{16(m+1)^2(\Delta f)^2\varepsilon}\right) \\
&\leqslant \frac{48 \cdot 16M^2(m+1)^4(\Delta f)^2}{\delta}\varepsilon \\
&\leqslant \frac{48(\Delta f)^2\delta}{2^{31}M^2(m+1)^{9/2}}\xi \\
&\leqslant \frac{\xi}{8}
\end{aligned}
$$

using $\Delta f \leqslant 2M$, $\delta \leqslant 1$, and $m+1 \geqslant 1$. Furthermore, by (3), we get that

$$16M(m+1)^{3/2} \cdot \frac{2^{17}M^3(m+1)^3}{D^2}\varepsilon = \frac{36 \cdot 2^{21}M^4(m+1)^{17/2}}{\delta^2}\varepsilon \leqslant \frac{\xi}{8}\,.$$

We therefore obtain the sought $\xi/4$ upper-bound and can conclude that

$$
\begin{aligned}
\int_0^1 (f_\eta(x; a(T), u(T)) - f^*(x))^2\mathrm{d}x &\leqslant \int_0^1 (f_\eta(x; a_\eta^*(u(T)), u(T)) - f^*(x))^2\mathrm{d}x \\
&\quad + 2|L_\eta(a(T), u(T)) - L_\eta(a_\eta^*(u(T)), u(T))| \\
&\leqslant \xi\,.
\end{aligned}
$$

## C  Experimental details and additional experiments

Our code is available at https://github.com/PierreMarion23/two-timescale-nn.

**Numerical illustration in the setting of Section 2.**    To obtain Figures 3 and 4, we use the parameters of Table 1. For Figure 5, we use the parameters of Table 2.

| Name | Value |
|---|---|
| $m$ | 20 |
| $\varepsilon$ | $2 \cdot 10^{-5}$ |
| $\eta$ | $4 \cdot 10^{-3}$ |
| $P$ | $1.8 \cdot 10^8$ |
| $h$ | $10^{-5}$ |
| Additive noise | Uniform on $[-1, 1]$ |

Table 1: Parameters of Figures 3 and 4.

| Name | Value |
|---|---|
| $m$ | 20 |
| $\varepsilon$ | 1 |
| $\eta$ | $4 \cdot 10^{-3}$ |
| $P$ | $10^6$ |
| $h$ | $10^{-5}$ |
| Additive noise | Uniform on $[-1, 1]$ |

Table 2: Parameters of Figure 5.

Our target function is defined by $f^* = 1$ on $[0., 0.2], [0.35, 0.5], [0.65, 0.8]$, $f^* = 2$ on $[0.5, 0.65]$ and $f^* = 4$ elsewhere. The activation function is a piecewise cubic polynomial defined by $x \mapsto \min(\max(4(x + 0.5)^3, 0), 0.5) + \min(\max(0.5 - 4(0.5 - x)^3, 0), 0.5)$.

We re-run the same SGD experiment as above twenty times, and plot in Figure 8 the average $L2$ distance to the target as a function of $\varepsilon$, averaging over the initialization randomness and SGD randomness. This confirms that, in our setting, the SGD is able to recover the target function in the two-timescale regime ($\varepsilon \ll 1$), but fails outside of the two-timescale regime ($\varepsilon = 1$). The transition between the two regimes seems to occur for $\varepsilon \approx 0.1$.

The number of iterations in Table 1 is much larger than the one in Table 2. There are two levels of analysis to explain this difference. The most straightforward reason is that the positions $u$ evolve at a speed $\varepsilon h$, which is much smaller in Table 1. However, one may note that it is not larger by a

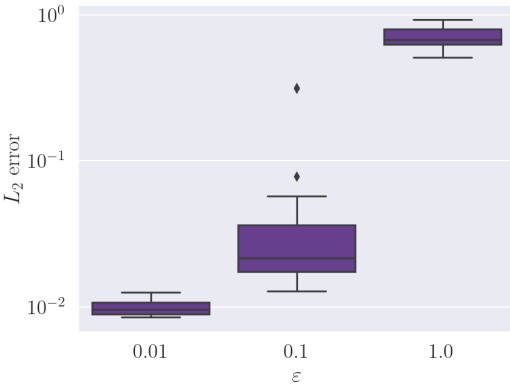

Figure 8: $L_2$ distance with the target as a function of $\varepsilon$ in the 1D experiment, with 20 repeats

factor $1/\varepsilon = 50,000$ but "only" by a factor $\simeq 200$. To understand why, we have to get more into the details of the dynamics, to understand the order of magnitude of the number of steps required before convergence. In the two-timescale regime, the limiting factor for convergence is the movement of the positions $u$. At each step, they move by an order of $\varepsilon h \simeq 10^{-10}$. The positions need to move on a scale of $5 \cdot 10^{-2}$ to align with the discontinuities of the target. Hence the required number of steps is $0.05/(2 \cdot 10^{-10}) \simeq 2 \cdot 10^8$. In the standard regime, on the contrary, the limiting factor for convergence is the movement of the weights $a$. They move by an order of $h \simeq 10^{-5}$ at each step, and they need to move by a distance of $\simeq 5$, necessitating $\simeq 5 \cdot 10^5$ steps to achieve convergence.

Finally, note that it is possible to increase $h$ in Table 1 while keeping the same behavior (in our experiment, $h$ is kept to the same value as in Table 2 in order to facilitate the comparison). More precisely, taking $h = 10^{-3}$ in Table 1 yields similar results while dividing the computational cost by 100.

**Higher dimensions.** In 2D, we use the parameters of Table 3 and 4. The positions are initialized uniformly over $[0, 1]$ and the weights uniformly over $[0, 3]$. The target is

$$f^*(x, y) = \mathbf{1}_{x \geqslant 0.3} + \mathbf{1}_{x \geqslant 0.5} + \mathbf{1}_{x \geqslant 0.7} + \mathbf{1}_{y \geqslant 0.3} + \mathbf{1}_{y \geqslant 0.5} + \mathbf{1}_{y \geqslant 0.7}.$$

The activation function is the same as in the one-dimensional case. We use SGD with batch size $10^3$.

| Name | Value |
|:---:|:---:|
| $m$ | 10 |
| $\varepsilon$ | 1 |
| $\eta$ | $10^{-2}$ |
| $P$ | 300 |
| $h$ | 1.0 |
| Additive noise | None |

Table 3: Parameters of Figure 6a.

| Name | Value |
|:---:|:---:|
| $m$ | 10 |
| $\varepsilon$ | $10^{-2}$ |
| $\eta$ | $10^{-2}$ |
| $P$ | $5 \cdot 10^3$ |
| $h$ | 1.0 |
| Additive noise | None |

Table 4: Parameters of Figure 6b.

As in the one-dimensional case, we re-run the experiment twenty times and report the results in Figure 9a.

We perform a similar experiment in 10D, using the same setup. The target is

$$f^*(x_1, \ldots, x_{10}) = \sum_{i=1}^{10} \mathbf{1}_{x_i \geqslant 0.3} + \mathbf{1}_{x_i \geqslant 0.5} + \mathbf{1}_{x_i \geqslant 0.7}.$$

We use SGD with batch size $10^4$. The results in terms of $L_2$ distance are reported in Figure 9b.

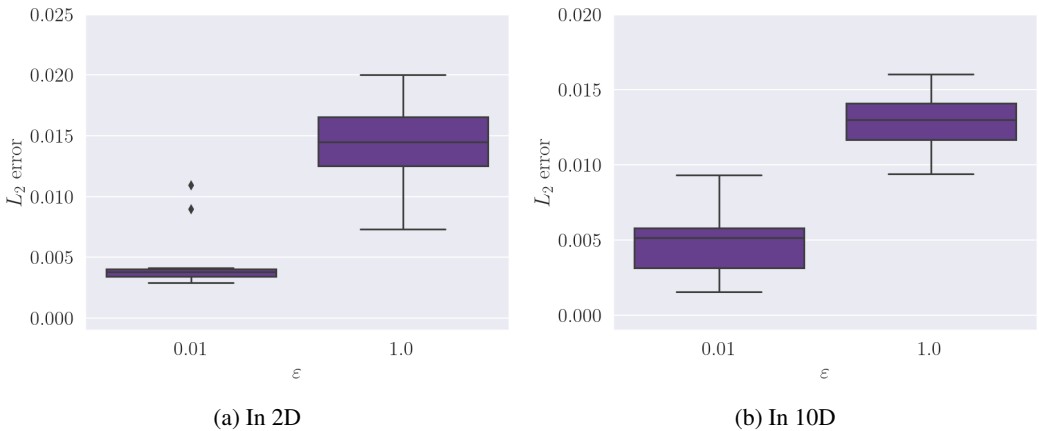

Figure 9: $L_2$ distance with the target as a function of $\varepsilon$ in the higher-dimensional experiment, with 20 repeats.

**ReLU networks.** The goal of this experiment is to use ReLU activations to approximate piecewise-affine targets in 1D. The target is a ReLU network with positions $u = [0.3, 0.5, 0.7]$ and weights $a = [1.0, -2.0, 3.0]$. We use the same setup as for the 2D experiment described above. The results are reported in Figure 10.

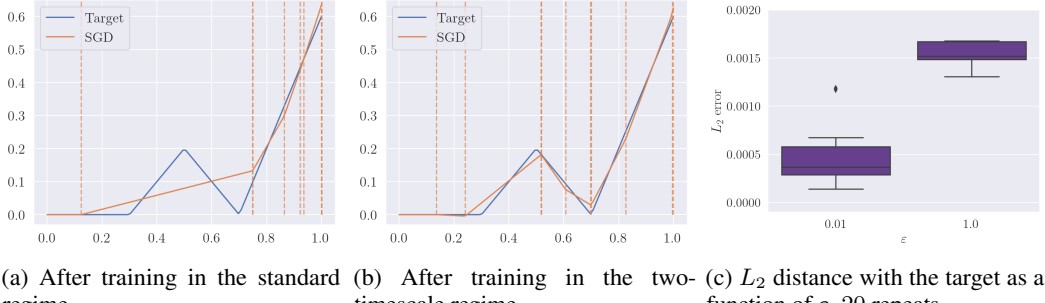

(a) After training in the standard regime

(b) After training in the two-timescale regime

(c) $L_2$ distance with the target as a function of $\varepsilon$, 20 repeats

Figure 10: Experiment with ReLU activation and piecewise-affine targets. In the standard regime, some kinks of the target are not covered by neurons. This is not the case in the two-timescale regime.

# D    Additional remarks

## D.1    Counter-example in the case of non-uniform initialization

It is easy to craft an example where gradient flow in the two-timescale limit does not converge to the global minimum if the positions of the neurons are not drawn uniformly at random at initialization. Take for instance the case where there are three pieces in the target, the leftmost piece at level 1, the second one at level $-1$ and the third one at level 0. Consider the case where all neurons are in the third piece at initialization. Then (in the two-timescale limit) the bias and the weights are all instantly equal to zero, which is the solution of the optimization problem in $a$. Looking at (10), this shows that the positions do not move, and thus that the neurons remain in this local minimum.

This example may seem artificial, but in fact, more generally, the case where two consecutive pieces are not covered by any neuron often corresponds to a local minimum of the two-timescale dynamics. This is why we require neurons on every piece at initialization to avoid falling in this local minimum.

## D.2    Ideas of proof for SGD

The proof of Theorem 1 consists in bounding the difference between the actual dynamics and the dynamics that we would have if the weights were given by $a_0^*(u(t))$, that is, by the weights in the

two-timescale limit with $\eta = 0$. This requires to bound two errors terms, one coming from the fact that $\eta > 0$, and one coming from the fact that $\varepsilon > 0$. By making $\eta$ and $\epsilon$ small enough, we ensure that both terms are small, which in turn allows to describe the trajectories of the weights and of the positions throughout the dynamics.

Moving from gradient flow to stochastic gradient descent induces two additional error terms: a discretization error, and a noise error. However, taking small enough step size allows to bound both error terms and thus ensures that (with high probability) the SGD dynamics are close to the gradient flow dynamics. For this reason, we believe that our proof structure (namely, bounding the difference with the two-timescale limit as in Proposition 4, then showing that this bound allows to describe the trajectories of the weights and of the positions as in Theorem 1) should adapt to the SGD case, at the condition that the step size is small enough.

