# OpenReview forum: "Leveraging the two-timescale regime to demonstrate convergence of neural networks"
_NeurIPS.cc/2023/Conference — NeurIPS 2023 poster_

### Official Review · Reviewer_VU9p · 2023-06-26

**Soundness:** 4 excellent
**Presentation:** 4 excellent
**Contribution:** 3 good
**Rating:** 7
**Confidence:** 3

**Summary:**

This paper studies the training of 2-layer neural networks and proposes a two-timescale limit/regime. In this
limit/regime, the learning rate of the first layer is much smaller than the learning rate of the second layer.
As a result, the training of the network can be viewed as training the first layer and performing linear regression
over the first layer outputs, which is simpler/more structured than training both layers at the same rate. To
demonstrate the usefulness of this strategy, this paper considers a toy model and shows that a 2-layer network can fit
a certain family of 1D piecewise constant functions. The authors also empirically show that SGD can fail to fit this
family of functions outside the two-timescale regime.

**Strengths:**

* The presentation is clear.
* This paper not only contains results in the $\varepsilon \to 0$ limit, but also non-asymptotic results for small but
  non-vanishing $\varepsilon$. The derivation in the $\eta \to 0, \varepsilon \to 0$ limit is clean (Sec. 4.2), and
  the asymptotic-to-non-asymptotic parts are also easy to follow.
* This paper reminds me of a technique that is gaining popularity in the theory community: training the network for
  one (large) step, freezing the first layer, and then performing linear regression in the second layer using the
  features learned in the first layer (cf. [AAM22], [DLS22]). The limitation of this technique is that since the
  first layer is fixed after the first layer, it lacks the ability to refine the learned features. I feel the
  strategy introduced in this paper is a potential remedy to this problem, as here we also have the linear regression
  part of the argument, but the first layer can be trained for multiple steps.

[AAM22] Abbe, Emmanuel, Enric Boix Adsera, and Theodor Misiakiewicz. “The Merged-Staircase Property: A Necessary and Nearly Sufficient Condition for SGD Learning of Sparse Functions on Two-Layer Neural Networks.” In Proceedings of Thirty Fifth Conference on Learning Theory, 4782–4887. PMLR, 2022. https://proceedings.mlr.press/v178/abbe22a.html.

[DLS22] Damian, Alex, Jason D. Lee, and Mahdi Soltanolkotabi. “Neural Networks Can Learn Representations with Gradient Descent.” arXiv, June 30, 2022. http://arxiv.org/abs/2206.15144.

**Weaknesses:**

* Almost all things in this paper are 1D and somewhat tailored to this specific piecewise constant function class.
  I wonder whether/how this can be generalized to higher dimensions, other network architectures, and more general
  function classes.
* It seems that the dynamics are still relatively local. That is, we need some neurons in each target interval at
  initialization, which may not be reasonable when the dimension is high.
* The authors should probably add some discussion on the training-for-one-large-step-type technique (see the Strengths
  part of the review).

**Questions:**

* How general is this strategy? Is it possible to apply this strategy to problems (even toy ones) with dimension higher
  than $1$ and more standard 2-layer networks?
* Can we relax the requirement on the initialization? For example, can we remove the requirement that each interval
  contains at least $6$ $u_j$?
  Intuitively, when some interval contains no neurons, it is still possible that some neurons from those fitted
  intervals can move to that interval and fit the target function on that interval.
  I know the current requirement is reasonable in your setting, I am asking this because I feel this type of global
  feature learning process is important when the dimension is high.

**Limitations:**

This is a theoretical work and, as far as I can see, has no potential negative societal impacts.

---

> ### Author Rebuttal · Authors · 2023-08-08
>
> > How general is this strategy? Is it possible to apply this strategy to problems (even toy ones) with dimension higher than 1
>  and more standard 2-layer networks?
>
> We have evidence showing that **the two-timescale strategy applies to other settings (higher-dimensional problems, ReLU networks for approximating piecewise-affine functions), see the common rebuttal**. Nevertheless, a full general understanding of the impact of the two-timescale regime is an open question, which is left for future work.
>
> > Can we relax the requirement on the initialization? For example, can we remove the requirement that each interval contains at least $6 u_j$ ? Intuitively, when some interval contains no neurons, it is still possible that some neurons from those fitted intervals can move to that interval and fit the target function on that interval. I know the current requirement is reasonable in your setting, I am asking this because I feel this type of global feature learning process is important when the dimension is high.
>
> We are rather pessimistic, for the following reason. It is reasonably easy to craft an example of a local minimum of the two-timescale problem (equation (5) of the paper) where a discontinuity of the target function is not covered by any neuron. **Without any assumption on the initialization, it is possible that the gradient flow dynamics may fall into this local minimum.** Avoiding such minima is a challenging next step which would certainly require additional ingredients. We will add this discussion (and exhibit such an example of a local minimum) in the next version of the paper.
>
> > The authors should probably add some discussion on the training-for-one-large-step-type technique (see the Strengths part of the review).
>
> Finally, we agree that the two-timescale regime can be seen as a refinement of layer-wise training, where we cyclically take one gradient step in the inner layer and many gradient steps in the outer layer. We thank you for the additional references; we will make sure to add them in the paper and discuss the connection.

---

> > ### Comment · Reviewer_VU9p · 2023-08-14
> >
> > Thanks for the response. I will keep my score.

---

### Official Review · Reviewer_83X8 · 2023-07-04

**Soundness:** 3 good
**Presentation:** 3 good
**Contribution:** 2 fair
**Rating:** 5
**Confidence:** 4

**Summary:**

This paper studied the problem of fitting a piecewise constant univariate function with a shallow neural network. Specifically, the authors consider a gradient flow with a time-scale difference between the dynamics of the first and the second layer weights. It is shown that the trained shallow network can be arbitrarily close to the target function in $\mathcal{L}_2$, as long as the weight update on the first layer is much slower than one on the second layer.

**Strengths:**

The two-time-scale regime seems novel in training neural networks. The results are well presented and their proofs are clearly explained.

**Weaknesses:**

Only a very special problem is studied.

1. The target function is univariate, piece-wise constant. This is very restrictive.

2. The loss is a population loss, thus the case of finite samples (fixed design or random samples) is not considered.

3. Network is quite different from those studied in other works. The activation function is a smoothed version of a stair function; the first layer is only parametrized by the bias at every neuron.

With this many assumptions/restrictions, even though the main results are well presented and explained, it is hard to see whether those observations can give insights into the usefulness of implementing two-time-scale training for practical networks. The author did not discuss the limitations of these assumptions, nor did they show how two-time-scale training can be used in practice, even empirically.

**Questions:**

See "Weaknesses"

**Limitations:**

See "Weaknesses"

---

> ### Author Rebuttal · Authors · 2023-08-08
>
> We agree with the reviewer that the main focus of this paper is on the study of a specific problem and that it does not readily lead to practical implications. Below, we detail why we still believe that our work contributes towards a theory of neural networks.
>
> **The theory of the optimization of neural networks is a notoriously hard problem; consequently, research on this topic makes progress at small steps.** For instance, the NeurIPS paper of Safran et al. (2022), to which we compare in detail, is similarly restricted. The other related works to which we compare in Section 3 are also restricted to specific settings: for instance, the mean-field setting is restricted to infinitely large neural networks and does not lead to quantitative convergence rates; the neural tangent kernel approach is also restricted to large neural networks and does not explain feature learning. Our work also has important restrictions, but is novel in providing a theory for neural networks with a moderate width and feature learning dynamics. Moreover, our description of the non-linear dynamics is remarkably precise and quantitative, compared to the other approaches.
>
> Furthermore, our paper underlies the **general applicability of the two-timescale technique** (see Section 4.1), beyond the specific one-dimensional illustration that we study in depth. The two-timescale regime is mathematically tractable if one can solve the two-timescale limit equations (5) of the paper. In this paper, we restrict ourselves to a simple setting for which the solution is simple (Section 4.2). This enables a fluid illustration of the central concept of this paper, the two-timescale regime. However, **any other setting for which Eqs. (5) can be solved could lead to a convergence analysis following the same lines.**
>
> More precisely, we have the following expectations for the generalizations that the reviewer suggests:
> 1. We believe that **the generalization to higher dimensional problems is possible, as shown by simulations**, although it is a theoretical challenge, which we reserve for future work. We develop this point in the **common part of the rebuttal**.
> 2. **Our results can be generalized to finite sample sizes** (say, for single-pass SGD) at the cost of greater technicalities in controlling the deviation from the two-timescale limit. This follows a perturbation analysis similar to the one of Section 5.2, with additional terms due to random sampling. This setting actually corresponds to the experiment described in lines 279-281, that shows that SGD indeed closely follows the behavior of the two-timescale limit. We will mention this possible generalization and add a rough sketch of proof in the next version of the paper.
> 3. We believe that **our results can be adapted to ReLU networks for approximating piecewise-affine functions, as shown by simulations** and as stated in the conclusion of the paper. Simulations show that the expected alignment of neurons with the discontinuities (of the derivative) still holds **(see common part of the rebuttal)**, although the mathematical derivations is more arduous.
>
> In dimension $1$, we also believe that **it should be possible to analyze more standard parametrizations of the inner layer** of the form
> $$f(x; a, u) = a_0 + \sum_{j=1}^m a_{j} \sigma_\eta (v_j x - u_{j}).$$
> However, this would make the parametrization of the “position” of neuron $j$, namely $u_j / v_j$, less natural, and thus would add technical difficulties to the proofs. We avoided this technicality for simplicity.

---

> > ### Comment · Reviewer_83X8 · 2023-08-14
> >
> > Thank the authors for the response. I personally still think the assumptions (1. and 3. in my original review) are very strong. Nonetheless, the authors provide additional numerical experiments in the rebuttal showing these assumptions might be relaxed, thus I raised the score.

---

> > > ### Author Response · Authors · 2023-08-15
> > >
> > > Thank you for your questions that contribute to improving the paper by adding a thorough discussion on possible generalizations, and for your willingness to raise the score.

---

### Official Review · Reviewer_JDX5 · 2023-07-07

**Soundness:** 3 good
**Presentation:** 3 good
**Contribution:** 3 good
**Rating:** 6
**Confidence:** 4

**Summary:**

In this paper, the authors considered the problem of learning piece-wise linear function in 1d using two-layer neural network. They considered gradient flow on mean-square loss with different learning rates for 2 layers (two-timescale). Specifically, the outer layer weights are moving much faster than the inner layer weights. The activation used in 2-layer network is similar to a rescale version of sigmoid activation. This paper shows that under proper choice of the parameters, GF converges to small loss within polynomial time in relevant parameters. Experiments are provided to support the results.

**Strengths:**

1.	Understanding the training dynamics and convergence of neural networks is an important problem.
2.	The paper is overall easy to follow and clearly written. The proof sketch is given so that the reader can understand the main proof idea.
3.	The idea of this two-timescale/two different learning rates in analyzing the training dynamics for neural networks seems to be interesting.


**Weaknesses:**

1.	The problem considered is only in 1d and it would be interesting to see if the analysis could be generalized to multi-dimension.

**Questions:**

1.	In the experiments, I was wondering if one could elaborate in Figure 5 that why ‘the dynamics create a zone with no neuron’. It seems to me that there are neurons near every discontinuity of $f^*$. Also, it seems in Figure 4(c) and Figure 5(c), the number of steps for training are different, what is the reason for that?

**Limitations:**

The limitation is discussed in the paper. This is a theoretical work and therefore no foresee negative societal impact.

---

> ### Author Rebuttal · Authors · 2023-08-08
>
> > The problem considered is only in 1d and it would be interesting to see if the analysis could be generalized to multi-dimension.
>
> This question is shared with the other reviewers and is adressed in the common rebuttal.
>
> > In the experiments, I was wondering if one could elaborate in Figure 5 that why ‘the dynamics create a zone with no neuron’. It seems to me that there are neurons near every discontinuity of $f^*$.
>
> In Figures 4 and 5, **the neurons are represented by vertical dashed lines** (the orange round markers do not represent neurons, they are only present for black&white and color blind lisibility). In Figure 5(c), the discontinuity of the target function at $x=0.5$ is not covered by a neuron: the neural network has a flat zone between $0.41$ and $0.6$, whereas it should jump at $0.5$ if it were optimal. This stands in sharp contrast with Figure 4(c) where neurons are evenly distributed on the whole interval. We will improve the readability of this figure in the next version.
>
> > Also, it seems in Figure 4(c) and Figure 5(c), the number of steps for training are different, what is the reason for that?
>
> It could seem like we are favouring the algorithm in the two-timescale regime by running it on a larger number of steps. In fact, it is not the case because **we ran both algorithms until convergence**; running the algorithm with $\varepsilon = 1$ for a larger number of steps would not lead to an improvement.
>
> Further, we can elaborate on the difference of steps to reach convergence in the two cases. There are two levels of analysis to explain this difference. The most straightforward reason is that the number of steps should scale more or less as the inverse of $\varepsilon$, since each step (in $u$) gets scaled down by $\varepsilon$ in the two-timescale regime. This explains why the number of steps is much larger in Figure 4.
>
> However, you may note that it is not larger by a factor $1/\varepsilon = 50{,}000$, but “only” by a factor $\simeq200$. To understand why, we have to get more into the details of the dynamics, to understand the order of magnitude of the number of steps required before convergence.
> In Figure 4, the limiting factor for convergence is the movement of the positions $u$. At each step, they move by an order of $\varepsilon h \simeq 10^{-10}$. The positions need to move on a scale of $5 \cdot 10^{-2}$ to align with the discontinuities of the target. Hence the required number of steps is $0.05 / (2 \cdot 10^{-10}) \simeq 2 \cdot 10^8$.
> In Figure 5, on the contrary, the limiting factor for convergence is the movement of the weights $a$. They move by an order of $h \simeq
>  10^{-5}$ at each step, and they need to move by a distance of $\simeq 5$, necessitating $\simeq 5 \cdot 10^5$ steps to achieve convergence.
>
> We will include a precision on this point in the camera-ready paper if accepted.

---

> > ### Comment · Reviewer_JDX5 · 2023-08-11
> >
> > Thanks for the response to address my question. I will keep my score.

---

### Official Review · Reviewer_WNiU · 2023-08-02

**Soundness:** 4 excellent
**Presentation:** 3 good
**Contribution:** 3 good
**Rating:** 6
**Confidence:** 4

**Summary:**

The paper studies the training dynamics of fitting a one hidden layer shallow network with heaviside activation to a piecewise ground truth function with one-dimensional input. It proves that gradient flow always recovers the ground truth in finite time with only mild over-parametrization.

**Strengths:**

This paper is well-written, and intuitions behind technical proofs are well-presented, and theoretical results are well-supported by numerical experiments. The theoretical result itself is a nice observation, despite being in the simple one-dimensional input case.

**Weaknesses:**

The paper only considers the one-dimensional case. Based on the proof techniques, it is not clear if it is extendable to high dimension, which is of ultimate interests in the deep learning theory community, since the derivation in section 4.2 would not hold any more. While it is understandable that such a result would be difficult to obtain in high dimension, the paper didn't present any experimental results in the high-dimensional case either.


**Questions:**

The paper would greatly benefit from a few comments on whether the authors would expect the same phenomenon to be observed in high dimension and what the difficulty would be. Some numerical experiments in the high dimensional case would be a great addition as well.

---

> ### Author Rebuttal · Authors · 2023-08-08
>
> Your question on the generalization to higher-dimensional problems, including numerical experiments, is shared with the other reviewers. It is thus addressed in the common rebuttal.

---

### Author Rebuttal · Authors · 2023-08-08

Dear reviewers,

We warmly thank you for your time and relevant comments, which will help us improve our work. If accepted, we will take into account your suggestions, making use of the additional page.

Since all reviewers raised the relevant question of the **applicability of our approach to more general settings**, in particular beyond dimension 1, we address it below, and leave the answers to other questions in individual responses.

Sincerely,

The authors

------

This paper introduces the two-timescale regime in a general setting (Section 4.1). The choice of the specific setting we analyze in detail (1D, piecewise-constant target, and bias-only first layer) is motivated by the simplicity to derive mathematically tractable expressions for dynamics of the two-timescale limit (Section 4.2). Relaxing these assumptions makes the analysis more complicated and is postponed to future work. Nevertheless, following the comments of the reviewers, we next present preliminary experimental results in dimension larger than 1 and for ReLU networks.

**Dimension larger than 1.** We first emphasize that piecewise constant functions are a complicated class in higher dimensions; for example, a neural network with sigmoid activations is able to learn such a function only under certain conditions, including that the shape of the constant regions should be polygons.
As a first step, on $\mathbb{R}^d$ $(d > 1)$, we consider multivariate neural networks of the form
$$f(x; a, u) = a_0 + \sum_{j=1}^m \sum_{k=1}^d a_{jk} \sigma_\eta (x_k - u_{jk}),$$
where the $j$-th neuron has a $d$-dimensional position $(u_{jk})$ and a $d$-dimensional weight $(a_{jk})_{1 \leq k \leq d}$. To ensure that the neural network can approximate the target piecewise constant function, we choose a target function of the same form. We then perform similar experiments as in Figures 4 and 5 in the main paper (and Figure 7 in the appendix), that is, to train the neural network with gradient descent, both in a two-timescale regime and in the standard regime.
We report the results in the attached PDF file for $d=2$ and $d=10$. In a nutshell, **the conclusions are similar to the paper: recovery can fail in the standard regime**, due to discontinuities of the target function that are not covered by neurons after training (or, in other words, the gradient descent converges to a local minimum). **By keeping the exact same setting but lowering epsilon to transition into the two-timescale regime, we obtain convergence to a global minimum.**

We expect that the corresponding mathematical derivation should be doable with similar tools as in our proof, at the cost of significant additional technicalities. Further generalizations beyond this somewhat artificial example are left for future work.

**ReLU networks.** In a similar spirit, it is also possible to consider the case of using **ReLU activations to approximate piecewise-affine targets**. The results are also reported in the attached PDF file (for dimension 1, but we could mix both extensions and also consider higher-dimensional problems in this case), and **a similar conclusion holds**. Here again, there are additional technical difficulties that complicate the proof, but we believe that there is no fundamental reason that our mathematical approach should not apply to this case.

These discussions and experiments will be added in the next version of the paper.

---

### Decision · Program_Chairs · 2023-09-21

**Decision:**

Accept (poster)

**Comment:**

The reviewers unanimously find the paper interesting and result worth publication. The area chair agrees with the evaluation after reading the discussion and the manuscript.